# Quantitative HBsAg versus HBV DNA in Predicting Significant Hepatitis Activity of HBeAg-Positive Chronic HBV Infection

**DOI:** 10.3390/jcm10235617

**Published:** 2021-11-29

**Authors:** Zhanqing Zhang, Wei Lu, Dong Zeng, Dan Huang, Weijia Lin, Li Yan, Yanling Feng

**Affiliations:** 1Department of Hepatobiliary Medicine, Shanghai Public Health Clinical Center of Fudan University, Shanghai 201508, China; luwei@shphc.org.cn (W.L.); huangdan@shphc.org.cn (D.H.); linweijia@shphc.org.cn (W.L.); yanli10389@shphc.org.cn (L.Y.); 2Department of Clinical Pathology, Shanghai Public Health Clinical Center of Fudan University, Shanghai 201508, China; zengdong@shphc.org.cn (D.Z.); fengyanling@shphc.org.cn (Y.F.)

**Keywords:** chronic hepatitis B virus infection, natural history, significant hepatitis activity, hepatitis B surface antigen, hepatitis B virus DNA

## Abstract

(1) Background: As specialparameters in predicting significant hepatitis activity of hepatitis B e antigen (HBeAg)-positive chronic hepatitis B virus (HBV) infection, the quantitative standard of HBV DNA has not been agreed and that of hepatitis B surface antigen(HBsAg) has not been formed. Our objective is to evaluate the validity of HBsAg and HBV DNA in predicting the significant hepatitis activity of HBeAg-positive patients. (2) Methods: A population of 516 patients with HBeAg-positive chronic HBV infection was enrolled. Serum ALT was measured using an Abbott Architect c16000 autoanalyzer; diagnoses of liver pathological grade and stage referred to the Scheuer standard. Three levels of significant hepatitis activity were preset, which were successively “ALT ≥ 20 IU/L or Grade > G1 or Stage > S1”, “ALT ≥ 30 IU/L or Grade > G1 or Stage > S1” and “ALT ≥ 40 IU/L or Grade > G1 or Stage > S1”. (3) Results: A subpopulation of 288 patients with possible high HBV replication was selected based on locally weighted scatterplot smoothing regression curves between ALT and HBsAg, HBeAg and HBV DNA. In the subpopulation with possible high HBV replication, areas under receiver operating characteristic curves of HBsAg for predicting the three levels of significant hepatitis activity were successively 0.868, 0.839 and 0.789, which were all significantly greater than those of HBV DNA, as those were successively 0.553, 0.550 and 0.574 (*p* = 0.0002, *p* < 0.0001 and *p* < 0.0001). With the standard of HBsAg ≤ 4.699 log_10_ IU/mL, the sensitivity and specificity of HBsAg for predicting the three levels of significant hepatitis activity were successively 75.81% and 81.82%, 79.23% and 78.57% and 80.82% and 67.44%. (4) Conclusion: Quantitative HBsAg instead of HBV DNA is valuable in predicting significant hepatitis activity of HBeAg-positive chronic HBV infection.

## 1. Introduction

Chronic hepatitis B virus (HBV) infection remains a major public health issue affecting human health [1]. Liver cirrhosis, hepatocellular carcinoma and liver decompensation caused by persistent or recurring significant hepatitis activity are the main adverse consequences of chronic HBV infection. Nucleoside/nucleotide antiviral drugs are the main measures to limit significant hepatitis activity and prevent the adverse consequences of chronic HBV infection [2,3,4]. Accurately identifying significant hepatitis activity of chronic HBV infection is a prerequisite for the rational management of patients and reasonable use of antiviral drugs.

With reference to serum hepatitis B e antigen (HBeAg) status and alanine transferase (ALT) and HBV DNA levels, the natural history of chronic HBV infection is generally divided into four consecutive but possibly reciprocating phases (See Section A.1 and Section A.2): (1) HBeAg-positive non-significant hepatitis activity, which is characterized by sustained normal ALT, fluctuating HBV DNA at high levels and sustained slight liver necro-inflammation with no or slight liver fibrosis; (2) HBeAg-positive significant hepatitis activity, which manifests as gradually or repeatedly elevated ALT, gradually or repeatedly decreased HBV DNA and gradually or repeatedly aggravated liver necro-inflammation with progressive liver fibrosis; (3) HBeAg-negative non-significant hepatitis activity, which is characterized by sustained normal ALT, fluctuating HBV DNA at low levels and slight liver necro-inflammation with gradually regressive liver fibrosis; and(4) HBeAg-negative significant hepatitis activity, which manifests as gradually or repeatedly elevated ALT, gradually or repeatedly increased HBV DNA and gradually or repeatedly aggravated liver necro-inflammationwith re-progressive liver fibrosis.

However, the upper limit of normal (ULN) for ALT is still controversial [2,3,4,5], and the delimitation of both ALT and HBV DNA levels in discriminating the natural history phases of chronic HBV infection is still not unified: (1) The rational reference value for ALT in predicting both HBeAg-positive and HBeAg-negative significant hepatitis activity is not conclusive yet [2,3,4], and the ability of ALT in predicting bothHBeAg-positive and HBeAg-negative liver fibrosis levels is limited [6,7,8,9]; (2) The rational cutoff value for HBV DNA in predicting HBeAg-positive significant hepatitis activity is not clear yet [2,3,4], and the validity of HBV DNA in predicting HBeAg-positive liver fibrosis levels is questioned [2,3,4,10,11].

The established quantitative serum markers of HBV include hepatitis B surface antigen (HBsAg), HBeAg, hepatitis B core-related antigen (HBcrAg), antibodies against hepatitis B core antigen (anti-HBc), HBV DNA and HBV RNA [11,12]. Among them, HBeAg and HBV RNA have not been standardized, and HBsAg has been promoted and applied [2,3,4]. In HBeAg-positive patients, HBsAg levels in the non-significant hepatitis activity phase were significantly higher than those in the significant hepatitis activity phase [13,14], and HBsAg showed good performance in predicting significant hepatitis activity, significant liver fibrosis with ALT less than 2 × ULN and liver cirrhosis without considering ALT levels [15,16,17]. However, in HBeAg-negative patients, HBsAg levels in the non-significant hepatitis activity phase were not necessarily significantly lower than those in the significant hepatitis activity phase [13,14], and HBsAg had no value in predicting significant hepatitis activity and liver fibrosis levels [15,16,17].

The quantitative standard for HBsAg in predicting the significant hepatitis activity phase of HBeAg-positive chronic HBV infection has not been formed [2,3,4]. HBsAg is produced by both HBV covalently closed circular DNA (cccDNA) and HBV DNA integrated into the host genome [18,19,20]. HBV DNA integration begins at the very early stage of chronic HBV infection [19,20], and the frequency of the integration in HBeAg-negative patients is significantly higher than that of HBeAg-positive patients [18,19,20]. Nevertheless, HBsAg in patients with HBeAg-positive chronic HBV infection was significantly positively correlated with HBV DNA [13,14,21]. In addition, the high levels of HBsAg associated with high other than medium or low HBV replication may contuse the host’s immune responses against HBV, which may be the direct key factor leading to non-significant hepatitis activity in patients with HBeAg-positive chronic HBV infection [15,22]. Therefore, the studies, whether by referring to the conventional phasing criteria of natural history that covers HBVDNA intrinsically related to the production of HBsAgor based on the overall population of HBeAg-positive stage that covers the subpopulation with low HBV replication, could not accurately obtain valuable information reflecting the performance of HBsAg in the prediction of HBeAg-positive significant hepatitis activity activity (See Section A.3, Section A.4, Section A.5, Section A.6, Section A.7 and Section A.8).

The purpose of this study was to delimit the subpopulation with possible high and possible low HBV replication in a population with HBeAg-positive chronic HBV infection, and on this basis, to evaluate the performance of HBsAg, HBeAg and HBV DNA for predicting significant HBeAg-positive hepatitis activity.

## 2. Materials and Methods

### 2.1. Study Population

HBeAg-positive patients with chronic HBV infection who were hospitalized and underwent liver biopsy in Shanghai Public Health Clinical Center of Fudan University from January 2017 to September 2020 were screened. A total of 516 patients with complete hospitalization records, detailed descriptions of liver pathology and matching laboratory data were enrolled for this study. Among them, 313 were males and 203 were females, and age ranged from 10 to 68 years with a median (interquartile range) of 34 (29–39) years; the proportions of patients with age <25 years and <30 years in males were 9.3% (29/313) and 30.0% (94/313), and in females were 6.4% (13/203) and 27.1% (55/203). Patients with the following conditions were excluded: coinfection with other hepatotropic viruses (hepatitis A, C, D and E virus), Epstein–Barr virus, cytomegalovirus; Schistosomiasis japonica liver disease, nonalcoholic/metabolic fatty liver disease (steatosis > 5%), drug-induced liver injury, excessive drinking (equivalent to ethanol, male >30 g/day, female >20 g/day), autoimmune diseases, endocrine and metabolic diseases, gallstones and bile duct stones, liver tumors and decompensated liver disease; antiviral therapy with interferon-α/peg-interferon-α or nucleosides/nucleotides, hepato-protective therapy with glycyrrhizates or oxymatrine/matrine or bicyclol/bifendate or traditional Chinese medicine prescriptions (including Chinese patent medicines) within 6 months before liver biopsy; length of biopsy specimen less than 1.0 cm or number of portal areas less than 6.

### 2.2. Laboratory Assays

Fasting venous blood was collected and serum was separated on the morning of liver biopsy. HBsAg and HBeAg were measured by microparticle enzyme immunoassay using an Abbott Architect i2000 automatic immunoassay system (Abbott Laboratories, Chicago, IL, USA) and auxiliary reagents. The detection range of HBsAg is 0.05 to 250 IU/mL, and a sample was diluted by 500 times and re-measured if HBsAg exceeded the upper limit of detection; the lower limit of detection of HBeAg is 1.0 SCO. HBV DNA was measured by PCR probe assay using a Roche LightCycler 480 qPCR system (Roche, Basel, Switzerland), the reagents were purchased from Sansure Biotech Inc. (Changsha, China) and the detection range of HBV DNA is 1 × 10^2^ to 2 × 10^9^ IU/mL. ALT was measured using an Abbott Architect c16000 automatic biochemical analysis system (Abbott Laboratories, Chicago, IL, USA) and auxiliary reagents. Based on related literature [2,3,4,5], three levels of ULNs for ALT were preset in this study, which were successively 20 IU/L, 30 IU/L and 40 IU/L, and “ALT ≥ ULNs” were defined as biochemically significant hepatitis activity.

### 2.3. Pathological Diagnoses

Percutaneous suction liver biopsies assisted by ultrasonography were performed using a 16G biopsy needle after written informed consent forms were signed. The biopsy specimens were immediately transferred into plastic tubes, quickly frozen and processed within 36 h. The quality assessments and pathological diagnoses of biopsy specimens were performed independently by one experienced pathologist, who was ignorant of the laboratory information. The pathological diagnoses of liver specimens referred to the Scheuer scoring system [23], in which the intensity of necro-inflammation is divided into five grades ranging from G0 to G4, and the degree of fibrosis and alteration of architecture is divided into five stages ranging from S0 to S4. In this study, “Grade > G1 or Stage > S1” was defined as pathologically significant hepatitis activity.

### 2.4. Statistical Analyses

MedCalc version 15.8 (MedCalc Software, Mariakerke, Belgium) was used for statistical analyses and graph productions. Locally weighted scatterplot smoothing (LOESS) regression analysis was used to explore the evolving trend between ALT and HBsAg, HBeAg and HBV DNA, and to delimit the subpopulations with possible high and possible low HBV replication. Mann–Whitney*U* non-parametric test was used to compare the differences in ALT, HBsAg, HBeAg and HBV DNA levels between subpopulations with possible high and with possible low HBV replication. Spearman rank correlation analysis was used to analyze the correlations between ALT, liver pathological grade and stage and HBsAg, HBeAg and HBV DNA. Fisher *Z* non-parametric test was used to compare the differences in Spearman correlation coefficients between ALT, liver pathological grade and stage and HBsAg, HBeAg and HBV DNA between each other. Receiver operating characteristic (ROC) curve was used to evaluate the performance of HBsAg, HBeAg and HBV DNA for predicting “biochemical or pathological” significant hepatitis activity in the subpopulation with possible high HBV replication and for predicting biochemically significant hepatitis activity in the subpopulation with possible low HBV replication. Dependent sample Hanley and McNeil *Z* nonparametric test was used to compare the differences in areas under ROC curves (AUCs) between HBsAg, HBeAg and HBV DNA for predicting significant hepatitis activity. *p* < 0.05 was defined as statistically significant.

## 3. Results

### 3.1. Delimitation of Subpopulation with Possible High and Possible Low HBV Replication

HBsAg, HBeAg and HBV DNA were used as independent variables, ALT was used as a dependent variable and LOESS regression analyses were performed with a span of 60%. The LOESS regression curves between ALT and HBsAg, HBeAg and HBV DNA are illustrated in Figure 1.

According to the LOESS regression curves, HBsAg, HBeAg and HBV DNA levels could be divided into two strata of high and medium–low levels: HBsAg > 4.250 and ≤4.250 log_10_ IU/mL, HBeAg > 2.875 and ≤2.875 log_10_ SCO and HBV DNA > 7.500 and ≤7.500 log_10_ IU/mL. The proportions of ALT ≥ 20 IU/L, ≥ 30 IU/L and ≥ 40 IU/L in patients with high levels of HBsAg were 92.0% (161/175), 79.4% (139/175) and 70.9% (124/175), with high levels of HBeAg were 93.6% (221/236), 84.7% (200/236) and 77.1% (182/236), and with high levels of HBV DNA were 94.4% (184/195), 85.1% (166/195) and 77.4% (151/195).

With the standard of the quantitative stratification of HBsAg, HBeAg and HBV DNA, the subpopulations with possible high HBV replication and possible low HBV replication were defined as “HBsAg > 4.250 log_10_ IU/mL or HBeAg > 2.875 log_10_ SCO or HBV DNA > 7.500 log_10_ IU/mL” and “HBsAg ≤ 4.250 log_10_ IU/mL and HBeAg ≤ 2.875 log_10_ SCO and HBV DNA ≤ 7.500 log_10_ IU/mL”. In the subpopulation with possible high HBV replication, the proportions of patients with high levels of HBsAg, HBeAg and HBV DNA were 60.8% (175/288), 81.9% (236/288) and 67.7% (195/288).

### 3.2. Demographic, Laboratory and Pathological Characteristics of Study Population

The demographic, laboratory and pathological characteristics of the overall population and of the subpopulations with possible high and possible low HBV replication are summarized in Table 1.

In the subpopulation with possible high HBV replication, the proportions of patients with age < 25 years and < 30 years in males were 8.4% (15/179) and 33.0% (59/179), and in females were 6.4% (7/109) and 24.8% (27/109). In the subpopulation with possible low HBV replication, the proportions of patients with age < 25 years and < 30 years in males were 10.4% (14/134) and 26.1% (35/134), and in females were 6.4% (6/94) and 29.8% (28/94).

### 3.3. Correlation between HBsAg, HBeAg and HBV DNA and ALT, Pathological Grade and Stage

The Spearman correlation coefficients between HBsAg, HBeAg and HBV DNA and ALT, pathological grade and stage in the overall population and in the subpopulations with possible high and possible low HBV replication are summarized in Table 2.

### 3.4. Performance of HBsAg, HBeAg and HBV DNA in Predicting Significant Hepatitis Activity

In the overall population, the AUCs of HBsAg, HBeAg and HBV DNA for predicting “ALT ≥ 20 IU/L or Grade > G1 or Stage > S1” were 0.660, 0.573 and 0.538, for predicting “ALT ≥ 30 IU/L or Grade > G1 or Stage > S1” were 0.682, 0.600 and 0.504, and for predicting “ALT ≥ 40 IU/L or Grade > G1 or Stage > S1” were 0.654, 0.580 and 0.522; among them, the AUC of HBsAg for predicting “ALT ≥ 20 IU/L or Grade > G1 or Stage > S1” was significantly greater than that of HBV DNA (*Z* = 2.143, *p* = 0.0321), the AUC of HBsAg and HBeAgfor predicting “ALT ≥ 30 IU/L or Grade > G1 or Stage > S1” was significantly greater than that of HBV DNA (*Z* = 4.588, *p* < 0.0001 and *Z* = 2.348, *p* = 0.0189), and the AUC of HBsAg for predicting “ALT ≥ 40 IU/L or Grade > G1 or Stage > S1” was significantly greater than that of HBeAg and HBV DNA (*Z* = 2.013, *p* = 0.0441 and *Z* = 4.280, *p* < 0.0001).

In the overall population, the AUCs of HBsAg, HBeAg and HBV DNA for predicting “ALT ≥ 20 IU/L” were 0.551, 0.569 and 0.661, for predicting “ALT ≥ 30 IU/L” were 0.533, 0.575 and 0.647, respectively, and for predicting “ALT ≥ 40 IU/L” were 0.507, 0.592 and 0.635; among them, the AUC of HBsAg and HBeAg for predicting “ALT ≥ 20 IU/L” was significantly less than that of HBV DNA (*Z* = 2.529, *p* = 0.0114 and *Z* = 1.997, *p* = 0.0459), the AUC of HBsAg and HBeAg for predicting “ALT ≥ 30 IU/L” was significantly less than that of HBV DNA (*Z* = 4.092, *p* < 0.0001 and *Z* = 2.347, *p* = 0.0189), and the AUC of HBsAg for predicting “ALT ≥ 40 IU/L” was significantly less than that of HBeAg and HBV DNA (*Z* = 3.119, *p* = 0.0018 and *Z* = 5.385, *p* < 0.0001).

In the subpopulations with possible high and possible low HBV replication, the ROC curves and AUCs of HBsAg, HBeAg and HBV DNA for predicting “ALT ≥ 20 IU/L or Grade > G1 or Stage > S1” and “ALT ≥ 20 IU/L”, “ALT ≥ 30 IU/L or Grade > G1 or Stage > S1” and “ALT ≥ 30 IU/L”, and “ALT ≥ 40 IU/L or Grade > G1 or Stage > S1” and “ALT ≥ 40 IU/L” are illustrated in Figure 2 and summarized in Table 3.

Based on the Youden index, an optimal cutoff was determined. With reference to the minimum difference between the specificity of HBsAg and HBeAg for predicting “ALT ≥ 20 IU/L or Grade > G1 or Stage > S1” and the sensitivity of HBsAg and HBeAg for predicting “ALT ≥ 40 IU/L or Grade > G1 or Stage > S1” in the subpopulation with possible high HBV replication, and between the specificity of HBV DNA and HBeAg for predicting “ALT ≥ 20 IU/L” and the sensitivity of HBV DNA and HBeAg for predicting “ALT ≥ 40 IU/L” in the subpopulation with possible low HBV replication, a practical cutoff that was easy to remember was chosen. The optimal cutoffs and practical cutoffs with corresponding diagnostic parameters are summarized in Table 3.

### 3.5. Reliability of HBsAg, HBeAg and HBV DNA in Predicting Liver Pathological States

The proportions of liver pathological grades and stages in patients with HBsAg > 4.699 log_10_ IU/mL or HBeAg > 3.154 log_10_ SCO or HBV DNA ≤ 4.362 log_10_ IU/mL or HBeAg ≤ 0.861 log_10_ SCO alone and in combination with ALT < 20 IU/L, < 30 IU/L or < 40 IU/L are summarized in Table 4.

## 4. Discussion

In this study, based on a subpopulation with possible high and possible low HBV replication, we explored the role of HBsAg, HBeAg and HBV DNA in the persistence and onset of HBeAg-positive chronic HBV infection, and with reference to the criteria of “the preset ULNs for ALT” and “Grade > G1 or Stage > S1”, we evaluated the performance of HBsAg, HBeAg and HBV DNA in predicting significant hepatitis activity of HBeAg-positive chronic HBV infection. We also chose the practical cutoffs of HBsAg and HBeAg in predicting significant hepatitis activity in a subpopulation with possible high HBV replication and of HBV DNA and HBeAg in predicting significant hepatitis activity in a subpopulation with possible low HBV replication, and evaluated the reliability of HBsAg, HBeAg or HBV DNA alone and in combination with ALT in predicting liver pathological grades and stages.

So far, the nomenclature of four natural history phases of chronic HBV infection has not been fully unified [2,3,4]. Conventionally, HBeAg-positive non-significant hepatitis activity is titled “immune tolerance”, which is still adopted by the Asian Pacific Association for the Study of the Liver (APASL) guidelines (2015) and the American Association for the Study of Liver Diseases (AASLD) guidelines (2018) [2,3,4]. In recent years, the naming of “immune tolerance” has been challenged [24,25,26,27,28]. In fact, most of the patients in the“immune tolerance” phase have slight liver necro-inflammation, with no or slight liver fibrosis progression [24,25,26,27,28]. All patients enrolled in this study, including “normal” ALT with “high levels” of HBV DNA, have varying degrees of liver necro-inflammation and fibrosis. Thereby, the European Association for the Study of the Liver (EASL) guidelines (2017) have denominated “immune tolerance” as “HBeAg-positive chronic HBV infection” [3]. However, “immune tolerance” remains an important immunological mechanism for the persistence of chronic HBV infection [29], and “HBeAg-positive chronic HBV infection” does not deny the existence of the“immune tolerance” mechanism [3].

The virological mechanisms of the persistence and onset of chronic HBV infection have not been thoroughly elucidated [29,30,31]. The scatter plots between ALT and HBsAg, HBeAg and HBV DNA of this study showed that the proportions of ALT ≥ 40 IU/L in patients with high levels of HBsAg, HBeAg and HBV DNA were all greater than 70%, which suggested that biochemically significant hepatitis activity was common in patients with high HBV replication. The correlation analyses between ALT and HBsAg, HBeAg and HBV DNA of this study also showed that ALT in the subpopulation with possible high HBV replication was significantly negatively correlated with HBsAg and HBeAg, but not with HBV DNA, and in the subpopulation with possible low HBV replication was significantly positively correlated with HBV DNA and HBeAg, but not with HBsAg. These suggested that medium to high levels of HBsAg and HBeAg, by inhibiting the immune response against HBV [29,30,31], may induce the persistence of chronic HBV infection, while low to medium levels of HBV DNA and HBeAg, by activating the immune response against HBV, may cause the onset of chronic HBV infection.

The results of this study also support the inferences on the virological pathogenesis of chronic HBV infection based on experimental studies: high and medium–low levels of serum HBsAg, by inducing immune exhaustion and immune ignorance [32,33,34], may play a role in maintaining the persistence of chronic HBV infection, while high and medium–low levels of serum HBeAg, by inducing immune exhaustion and immune activation, may play a role in maintaining the persistence and in causing the onset of chronic HBV infection [35]. The levels of serum HBV DNA may reflect the levels of transcription and translation of HBV genes in liver cells. The high and medium–low levels of expression of intrahepatic HBcAg and HBeAg and medium–low levels of expression of serum HBeAg may all be the main targets of the host immune attack. The facultative effects of serum HBeAg-induced immune exhaustion and immune activation may be related to its successful short-sighted evolution [35].

To date, there is no consensus on ULNs for ALT, which can be used to predict significant hepatitis activity in chronic HBV infection [2,3,4]. The APASL guidelines (2015) and the EASL guidelines (2017) recommend the use of traditional normal references with ALT < 40 IU/L [2,3], and the AASLD guidelines (2018) recommend the use of novel normal references with ALT ≤ 35 IU/L for males and ≤ 25 IU/L for females [4]. However, the liver histological criteria based on Scheuer, Ludwig, Ishak and METAVIR scoring systems, which are widely used at present, have a high consistency in delimiting the pathological grades and stages of chronic hepatitis [36]. In this study, we preset three levels of ULNs with ALT ≥ 20 IU/L, ≥ 30 IU/L and ≥ 40 IU/L as criteria for biochemically significant hepatitis activity, and used “Grade > G1 or Stage > S1” of the Scheuer scoring system as criteria for pathologically significant hepatitis activity.

In patients with HBeAg-positive chronic HBV infection, non-significant hepatitis activity is not only related to high levels but also low levels of HBV replication. Among them, non-significant hepatitis activity related to high levels of HBV replication belongs to “persistent inactivity” of the“immune tolerance” phase, which is manifested as non-significant serum biochemical abnormalities and non-significant liver histological changes, while non-significant hepatitis activity related to low levels of HBV replication belongs to “transient inactivity” of the “immune activation” phase, which is manifested as non-significant serum biochemical abnormalities and unconfined liver histological changes (See Section A.8). Therefore, in this study, significant hepatitis activity of the subpopulation with possible high HBV replication was defined as “ALT ≥ ULNs or Grade > G1 or Stage > S1”, and with possible low HBV replication was defined as “ALT ≥ ULNs”. The ROC curve analyses of this study showed that, in the overall population, the AUCs of HBsAg, HBeAg and HBV DNA for predicting significant hepatitis activity were all less than 0.70; among them, the AUCs of HBsAg for predicting “ALT ≥ ULNs or Grade > G1 or Stage > S1” were all greater and all less than those of HBV DNA. The ROC curve analyses of this study also showed that, in the subpopulation with possible high HBV replication, the AUCs of HBsAg and HBeAg for predicting “ALT ≥ ULNs or Grade > G1 or Stage > S1” were all greater than 0.75 and 0.70, which were all significantly greater than those of HBV DNA as those were not valuable for predicting “ALT ≥ ULNs or pathological grade > G1 or stage > S1”; in contrast, in the subpopulation with possible low HBV replication, the AUCs of HBV DNA and HBeAg for predicting “ALT ≥ ULNs” were all greater than 0.75 and 0.70, which were all significantly greater than those of HBsAg as those were not valuable for predicting “ALT ≥ ULNs”. These suggested that, in patients with HBeAg-positive chronic HBV infection, HBsAg and HBV DNA are a reliable indicator in predicting the first significant hepatitis activity or “immune activation” and second significant hepatitis activity or “immune reactivation”, and HBeAg is the collaborative indicator of HBsAg and HBV DNA in predicting the first and second significant hepatitis activity; the delimitation of the subpopulations with possible high and possible low HBV replication has important value for reasonably and effectively evaluating the performance of HBsAg, HBeAg and HBV DNA for predicting significant hepatitis activity.

In this study, we chose the practical cutoffs of HBsAg and HBeAg and of HBV DNA and HBeAg, which were convenient for clinical application, for predicting significant hepatitis activity in subpopulations with possible high and low HBV replication. Based on information of this study population, in patients with ALT ≥ 20 IU/L, ≥ 30 IU/L and ≥ 40 IU/L, the proportions of liver pathological stage > S1 and > S2 were 44.4% and 11.2%, 47.3% and 17.6% and 47.1% and 21.0%. In patients with HBsAg > 4.699 log_10_ IU/mL, the proportions of liver pathological stage > S1 and > S2 were 31.2% and 1.3%, and in combination with ALT ≥ 40 IU/L, the proportions of liver pathological stage > S1 and > S2 were 17.1% and 0.0%. However, in patients with HBV DNA ≤ 4.362 log_10_ IU/mL, the proportions of liver pathological stage > S1 and > S2 were 65.8% and 38.4%, and in combination with ALT ≥ 20 IU/L, the proportions of liver pathological stage > S1 and > S2 were 68.8% and 25.0%. These suggested that the ability of ALT based on the three preset ULNs and of HBV DNA with reference to the practical cutoff alone and in combination with ALT for excluding extensive liver fibrosis and cirrhosis is all very limited; however, the performances of HBsAg with the standard of the practical cutoff alone and in combination with ALT for excluding extensive liver fibrosis and cirrhosis are both almost excellent.

This study had some limitations. First, the study population was all patients undergoing liver biopsy, and most of the patients were adults, so the selection bias of patients cannot be completely excluded. Second, the sample size was relatively small, and the gender differences in the normal references of ALT were not considered, so the findings of this study may need to be further confirmed in different genders by expanding the sample size. Third, this study was a retrospective cross-sectional study, and its findings may also need to be further verified by retrospective or prospective follow-up studies.

## 5. Conclusions

In the subpopulation with possible high HBV replication, HBsAg and HBeAg instead of HBV DNA were significantly negatively correlated with ALT, and had very good and good performance in predicting “biochemical or pathological” significant hepatitis activity; in the subpopulation with possible low HBV replication, HBV DNA and HBeAg instead of HBsAg were significantly positively correlated with ALT, and had very good and good performance in predicting biochemically significant hepatitis activity. With the standard of HBsAg > 4.699 log_10_ IU/mL and of ALT < 40 IU/L, HBsAg alone and in combination with ALT had excellent performance in excluding HBeAg-positive extensive liver fibrosis and cirrhosis of HBeAg-positive chronic HBV infection.

## Figures and Tables

**Figure 1 jcm-10-05617-f001:**
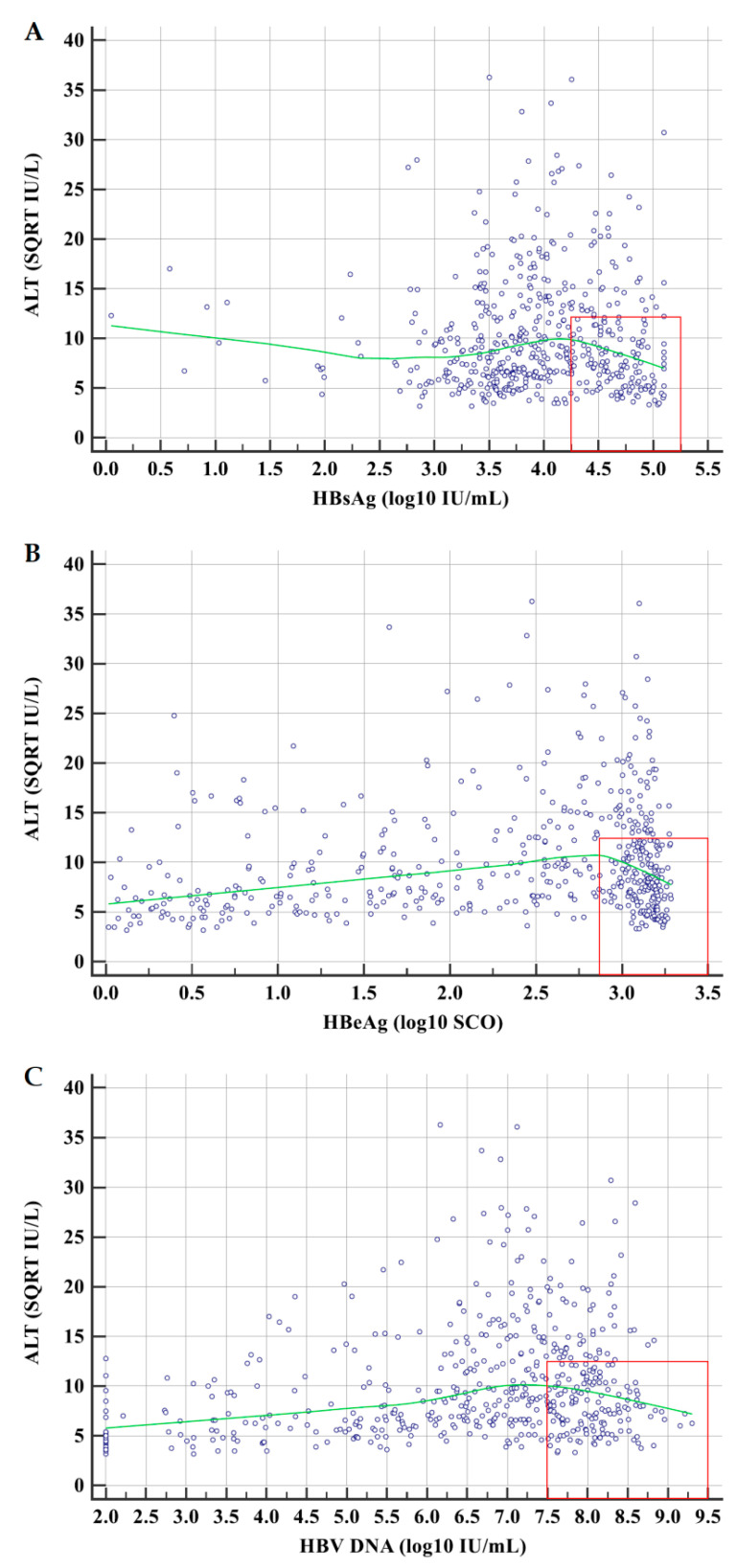
Scatter plots and LOESS regression curves between ALT and HBsAg (**A**), HBeAg (**B**) and HBV DNA (**C**). Abbreviations: ALT, alanine transferase; HBsAg, hepatitis B surface antigen; HBeAg, hepatitis B e antigen; HBV DNA, hepatitis B virus DNA; SQRT, square root; log_10_, logarithm with base 10. The red boxes are used to delineate high levels of HBsAg, HBeAg and HBV DNA.

**Figure 2 jcm-10-05617-f002:**
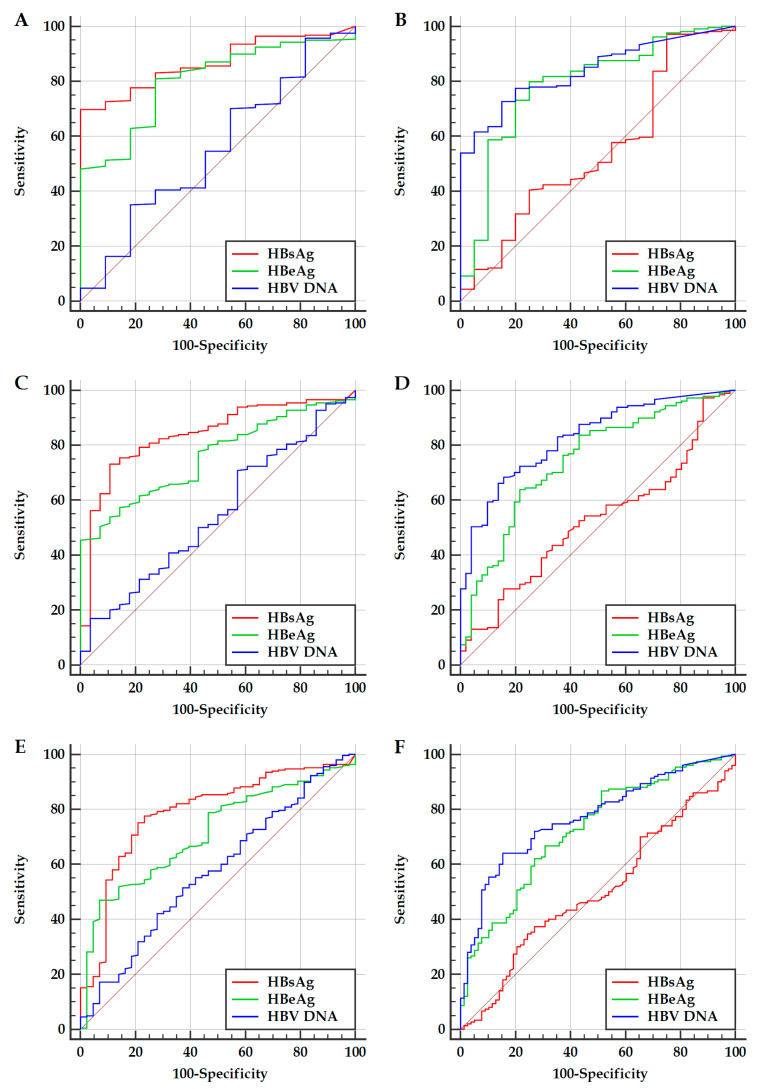
The ROC curves of HBsAg, HBeAg and HBV DNA for predicting significant hepatitis activity in subpopulation with possible high and low HBV replication. Abbreviations: ROC, receiver operating characteristic; HBsAg, hepatitis B surface antigen; HBeAg, hepatitis B e antigen; HBV DNA, hepatitis B virus DNA; ALT, alanine transferase. (**A**,**C**,**E**): Predicting “ALT ≥ 20 IU/L or Grade > G1 or Stage > S1” (**A**), “ALT ≥ 30 IU/L or Grade > G1 or Stage > S1” (**C**) and “ALT ≥ 40 IU/L or Grade > G1 or Stage > S1” (**E**) in patients with possible high HBV replication; (**B**,**D**,**F**): Predicting “ALT ≥ 20 IU/L” (**B**), “ALT ≥ 30 IU/L” (**D**) and “ALT ≥ 40 IU/L” (**F**) in patients with possible low HBV replication.

**Table 1 jcm-10-05617-t001:** Demographic, laboratory and pathological characteristics of study population.

Variable ^#^	Overall		Possible High HBV Replication		Possible Low HBV Replication		*χ^2^*^a^*Z*^b^ *	*p* *
	n	Summary Statistics	n	Summary Statistics	n	Summary Statistics		
Gender, male:female	516	313:203	288	179:109	228	134:94	0.608 ^a^	0.4354
Age, median (IQR)	516	34.0 (29.0–39.5)	288	32.0 (29.0–38.0)	228	35.0 (29.0–41.0)	2.731 ^b^	0.0063
ALT, median (IQR)	516	72.50 (38.00–164.00)	288	86.50 (44.00–180.00)	228	54.50 (32.00–134.50)	3.527 ^b^	0.0004
AST, median (IQR)	516	46.00 (28.00–87.00)	288	49.00 (30.00–92.50)	228	42.50 (27.00–73.00)	1.946 ^b^	0.0516
HBsAg, median (IQR)	516	3.949 (3.531–4.482)	288	4.431 (4.021–4.711)	228	3.535 (3.218–3.776)	16.221 ^b^	<0.0001
HBeAg, median (IQR)	516	2.758 (1.618–3.113)	288	3.100 (2.980–3.175)	228	1.585 (0.725–2.310)	17.572 ^b^	<0.0001
HBV DNA, median (IQR)	516	7.105 (5.693–7.853)	288	7.753 (7.264–8.152)	228	5.707 (3.932–6.598)	16.762 ^b^	<0.0001
Grade, G1:G2:G3:G4	516	268:190:58:0	288	171:95:22:0	228	97:95:36:0	17.066 ^a^	0.0002
Stage, S0–1:S2:S3:S4	516	211:160:65:80	288	151:91:24:22	228	60:69:41:58	56.708 ^a^	<0.0001

Abbreviations: ALT, alanine transferase; HBsAg, hepatitis B surface antigen; HBeAg, hepatitis B e antigen; HBV DNA, hepatitis B virus DNA; Grade, pathological grade; Stage, pathological stage; IQR, interquartile range. ^#^ Units of measurement: Age, years; ALT, IU/L; AST, IU/L; HBsAg, log_10_ IU/mL; HBeAg, log_10_ SCO; HBV DNA, log_10_ IU/mL. * Possible high HBV replication versus Possible low HBV replication: ^a^ Chi-squared test; ^b^ Mann–Whitney*U* test.

**Table 2 jcm-10-05617-t002:** Correlation between HBsAg, HBeAg and HBV DNA and ALT, liver pathological grade and stage.

Correlated Variables		Overall			Possible High HBV Replication			Possible Low HBV Replication			*Z* *	*p* *
Variable y	Variable x	n	*r_s_*	*p*	n	*r_s_*	*p*	n	*r_s_*	*p*		
ALT	HBsAg	516	0.004	0.9232	288	−0.303 ^a^	<0.0001	228	0.029 ^g,h^	0.6655	3.833	0.0001
ALT	HBeAg	516	0.131	0.0030	288	−0.268 ^b^	<0.0001	228	0.427 ^g^	<0.0001	8.196	<0.0001
ALT	HBV DNA	516	0.211	<0.0001	288	−0.109 ^a,b^	0.0639	228	0.530 ^h^	<0.0001	7.845	<0.0001
Grade	HBsAg	516	−0.291	<0.0001	288	−0.351 ^c^	<0.0001	228	−0.087^i,j^	0.1898	3.133	0.0017
Grade	HBeAg	516	−0.250	<0.0001	288	−0.375 ^d^	<0.0001	228	0.099 ^i^	0.1348	5.534	<0.0001
Grade	HBV DNA	516	−0.104	0.0182	288	−0.102 ^c,d^	0.0837	228	0.207 ^j^	0.0017	3.503	0.0005
Stage	HBsAg	516	−0.384	<0.0001	288	−0.303 ^e^	<0.0001	228	−0.135 ^k^	0.0414	1.985	0.0472
Stage	HBeAg	516	−0.373	<0.0001	288	−0.361 ^f^	<0.0001	228	0.019	0.7714	4.452	<0.0001
Stage	HBV DNA	516	−0.241	<0.0001	288	−0.089 ^e,f^	0.1311	228	0.062 ^k^	0.3542	1.697	0.0897

Abbreviations: ALT, alanine transferase; HBsAg, hepatitis B surface antigen; HBeAg, hepatitis B e antigen; HBV DNA, hepatitis B virus DNA; Grade, pathological grade; Stage, pathological stage. * Possible high HBV replication versus Possible low HBV replication: Fisher *Z* test. ^a–k^ Fisher *Z* test: ^a^, *Z* = 2.428, *p* = 0.0152; ^b^, *Z* = 1.973, *p* = 0.0485; ^c^, *Z* = 3.154, *p* = 0.0016; ^d^, *Z* = 3.484, *p* = 0.0005; ^e^, *Z* = 2.669, *p* = 0.0076; ^f^, *Z* = 3.448, *p* = 0.0006; ^g^, *Z* = 4.531, *p* < 0.0001; ^h^, *Z* = 5.952, *p* < 0.0001; ^i^, *Z* = 1.979, *p* = 0.0479; ^j^, *Z* = 3.153, *p* = 0.0016; ^k^, *Z* = 2.099, *p* = 0.0358.

**Table 3 jcm-10-05617-t003:** Performance of HBsAg, HBeAg and HBV DNA in predicting significant hepatitis activity.

**Predicting “ALT ≥ 20 IU/L or Grade > G1 or Stage > S1” in Subpopulation with Possible High HBV Replication**											
**Variable**	**AUC**	**SE**	** *Z* **	** *p* **	**Cutoff ***	**Sen (%)**	**Spe (%)**	**pLR**	**nLR**	**pPV (%)**	**nPV (%)**
HBsAg	0.868 ^a^	0.0332	11.087	<0.0001	≤4.615 ^Ⅰ^	69.68	100.00	+∞	0.30	100.0	11.6
					≤4.699 ^Ⅱ^	75.81	81.82	4.17	0.30	99.1	11.8
HBeAg	0.802 ^b^	0.0530	5.695	<0.0001	≤3.187 ^Ⅰ^	80.87	72.73	2.97	0.26	98.7	13.1
					≤3.154 ^Ⅱ^	70.40	72.73	2.58	0.41	98.5	8.9
HBV DNA	0.553 ^a,b^	0.0903	0.590	0.5549							
**Predicting “ALT ≥ 30 IU/L or Grade > G1 or Stage > S1” in Subpopulation with Possible High HBV Replication**											
**Variable**	**AUC**	**SE**	** *Z* **	** *p* **	**Cutoff ***	**Sen (%)**	**Spe (%)**	**pLR**	**nLR**	**pPV (%)**	**nPV (%)**
HBsAg	0.839 ^c^	0.0355	9.540	<0.0001	≤4.615 ^Ⅰ^	73.08	89.29	6.82	0.30	98.4	26.3
					≤4.699 ^Ⅱ^	79.23	78.57	3.70	0.26	97.2	28.9
HBeAg	0.758 ^d^	0.0363	7.099	<0.0001	≤3.062 ^Ⅰ^	45.38	100.00	+∞	0.55	100.0	16.5
					≤3.154 ^Ⅱ^	71.54	57.14	1.67	0.50	93.9	17.8
HBV DNA	0.550 ^c,d^	0.0551	0.911	0.3621							
**Predicting “ALT ≥ 40 IU/L or Grade > G1 or Stage > S1” in Subpopulation with Possible High HBV Replication**											
**Variable**	**AUC**	**SE**	** *Z* **	** *p* **	**Cutoff ***	**Sen (%)**	**Spe (%)**	**pLR**	**nLR**	**pPV (%)**	**nPV (%)**
HBsAg	0.789 ^e^	0.0372	7.766	<0.0001	≤4.663 ^Ⅰ^	77.55	76.74	3.33	0.29	95.0	37.5
					≤4.699 ^Ⅱ^	80.82	67.44	2.48	0.28	93.4	38.2
HBeAg	0.713 ^f^	0.0376	5.684	<0.0001	≤3.062 ^Ⅰ^	46.94	93.02	6.73	0.57	97.5	23.5
					≤3.154 ^Ⅱ^	72.65	53.49	1.56	0.51	89.9	25.6
HBV DNA	0.574 ^e,f^	0.0470	1.571	0.1162							
**Predicting “ALT ≥ 20 IU/L” in Subpopulation with Possible Low HBV Replication**											
**Variable**	**AUC**	**SE**	** *Z* **	** *p* **	**Cutoff ***	**Sen (%)**	**Spe (%)**	**pLR**	**nLR**	**pPV (%)**	**nPV (%)**
HBsAg	0.549 ^g,h^	0.0710	0.671	0.5024							
HBeAg	0.789 ^g^	0.0562	5.030	<0.0001	>0.718 ^Ⅰ^	79.81	75.00	3.19	0.27	97.1	26.3
					>0.861 ^Ⅱ^	73.08	80.00	3.65	0.34	97.4	22.2
HBV DNA	0.838 ^h^	0.0341	9.823	<0.0001	>4.756 ^Ⅰ^	72.60	85.00	4.84	0.32	98.1	23.0
					>4.362 ^Ⅱ^	74.52	80.00	3.73	0.32	97.5	23.2
**Predicting “ALT ≥ 30 IU/L” in Subpopulation with Possible Low HBV Replication**											
**Variable**	**AUC**	**SE**	** *Z* **	** *p* **	**Cutoff ***	**Sen (%)**	**Spe (%)**	**pLR**	**nLR**	**pPV (%)**	**nPV (%)**
HBsAg	0.526 ^i,j^	0.0437	0.582	0.5609							
HBeAg	0.745 ^i,k^	0.0391	6.244	<0.0001	>1.398 ^Ⅰ^	63.84	78.43	2.96	0.46	91.1	38.5
					>0.861 ^Ⅱ^	76.84	60.78	1.96	0.38	87.2	43.1
HBV DNA	0.827 ^j,k^	0.0300	10.858	<0.0001	>5.418 ^Ⅰ^	68.36	84.31	4.36	0.38	93.8	43.4
					>4.362 ^Ⅱ^	79.66	64.71	2.26	0.31	88.7	47.8
**Predicting “ALT ≥ 40 IU/L” in Subpopulation with Possible Low HBV Replication**											
**Variable**	**AUC**	**SE**	** *Z* **	** *p* **	**Cutoff ***	**Sen (%)**	**Spe (%)**	**pLR**	**nLR**	**pPV (%)**	**nPV (%)**
HBsAg	0.502 ^l,m^	0.0399	0.050	0.9600							
HBeAg	0.724 ^l^	0.0347	6.441	<0.0001	>1.398 ^Ⅰ^	66.67	69.23	2.17	0.48	80.6	51.9
					>0.861 ^Ⅱ^	78.67	51.28	1.61	0.42	75.6	55.6
HBV DNA	0.770 ^m^	0.0314	8.592	<0.0001	>5.860 ^Ⅰ^	64.00	84.62	4.16	0.43	88.9	55.0
					>4.362 ^Ⅱ^	80.00	50.00	1.60	0.40	75.5	56.5

Abbreviations: ROC, receiver operating characteristic; AUC, area under ROC curve; SE, standard error; HBsAg, hepatitis B surface antigen; HBeAg, hepatitis B e antigen; HBV DNA, hepatitis B virus DNA; ALT, alanine transferase; Grade, pathological grade; Stage, pathological stage; Sen, sensitivity; Spe, specificity; pLR, positive likelihood ratio; nLR, negative likelihood ratio; pPV, positive predictive value; nPV, negative predictive value.* Units of measurement: HBsAg, log_10_ IU/mL; HBeAg, log_10_ SCO; HBV DNA, log_10_ IU/mL. ^Ⅰ^ Optimal cutoff; ^Ⅱ^ Practical cutoff. ^a–l^ Dependent sample Hanley and McNeil nonparametric test: ^a^, *Z* = 3.779, *p* = 0.0002; ^b^, *Z* = 2.697, *p* = 0.0070; ^c^, *Z* = 5.131, *p* < 0.0001; ^d^, *Z* = 3.479, *p* = 0.0005; ^e^, *Z* = 4.385, *p* < 0.0001; ^f^, *Z* = 2.557, *p* = 0.0106; ^g^, *Z* = 2.705, *p* = 0.0068; ^h^, *Z* = 3.917, *p* = 0.0001; ^i^, *Z* = 3.898, *p* = 0.0001; ^j^, *Z* = 6.468, *p* < 0.0001; ^k^, *Z* = 2.128, *p* = 0.0333; ^l^, *Z* = 4.310, *p* < 0.0001; ^m^, *Z* = 6.010, *p* < 0.0001.

**Table 4 jcm-10-05617-t004:** Reliability of HBsAg, HBeAg or HBV DNA alone and in combination with ALT in predicting pathological states.

Variable ^#^	n	Proportion of Pathological Grades			Proportion of Pathological Stages			
		G1, % (n)	G2, % (n)	G3, % (n)	S0–1, % (n)	S2, % (n)	S3, % (n)	S4, % (n)
ALT < 20	36	83.3 (30)	13.9 (5)	2.8 (1)	55.6 (20)	33.3 (12)	5.6 (2)	5.6 (2)
ALT < 30	91	76.9 (70)	19.8 (18)	3.3 (3)	52.7 (48)	29.7 (27)	7.7 (7)	9.9 (9)
ALT < 40	138	73.9 (102)	20.3 (28)	5.8 (8)	52.9 (73)	26.1 (36)	9.4 (13)	11.6 (16)
HBsAg > 4.699	77	84.4 (65)	14.3 (11)	1.3 (1)	68.8 (53)	29.9 (23)	1.3 (1)	0.0 (0)
HBsAg > 4.699 and ALT < 20	10	100.0 (10)	0.0 (0)	0.0 (0)	90.0 (9)	10.0 (1)	0.0 (0)	0.0 (0)
HBsAg > 4.699 and ALT < 30	27	96.3 (26)	3.7 (1)	0.0 (0)	81.5 (22)	18.5 (5)	0.0 (0)	0.0 (0)
HBsAg > 4.699 and ALT < 40	35	97.1 (34)	2.9 (1)	0.0 (0)	82.9 (29)	17.1 (6)	0.0 (0)	0.0 (0)
HBeAg > 3.154	90	75.6 (68)	22.2 (20)	2.2 (2)	68.9 (62)	25.6 (23)	2.2 (2)	3.3 (3)
HBeAg > 3.154 and ALT < 20	10	90.0 (9)	10.0 (1)	0.0 (0)	90.0 (9)	10.0 (1)	0.0 (0)	0.0 (0)
HBeAg > 3.154 and ALT < 30	22	95.5 (21)	4.5 (1)	0.0 (0)	77.3 (17)	22.7 (5)	0.0 (0)	0.0 (0)
HBeAg > 3.154 and ALT < 40	29	96.6 (28)	3.4 (1)	0.0 (0)	82.8 (24)	17.2 (5)	0.0 (0)	0.0 (0)
HBV DNA ≤ 4.362	73	54.8 (40)	32.9 (24)	12.3 (9)	34.2 (25)	27.4 (20)	11.0 (8)	27.4 (20)
HBV DNA ≤ 4.362 and ALT < 20	16	75.0 (12)	18.8 (3)	6.2 (1)	31.2 (5)	43.7 (7)	12.5 (2)	12.5 (2)
HBV DNA ≤ 4.362 and ALT < 30	33	63.6 (21)	30.3 (10)	6.1 (2)	36.4 (12)	36.4 (12)	9.1 (3)	18.2 (6)
HBV DNA ≤ 4.362 and ALT < 40	39	61.5 (24)	28.2 (11)	10.3 (4)	35.9 (14)	33.3 (13)	10.3 (4)	20.5 (8)
HBeAg ≤ 0.861	76	47.4 (36)	35.5 (27)	17.1 (13)	30.3 (23)	31.6 (24)	13.2 (10)	25.0 (19)
HBeAg ≤ 0.861 and ALT < 20	16	81.2 (13)	12.5 (2)	6.2 (1)	37.5 (6)	43.7 (7)	6.2 (1)	12.5 (2)
HBeAg ≤ 0.861 and ALT < 30	31	67.7 (21)	25.8 (8)	6.5 (2)	38.7 (12)	32.3 (10)	12.9 (4)	16.1 (5)
HBeAg ≤ 0.861 and ALT < 40	41	65.9 (27)	24.4 (10)	9.8 (4)	39.0 (16)	31.7 (13)	12.2 (5)	17.1 (7)

Abbreviations: ALT, alanine transferase; HBsAg, hepatitis B surface antigen; HBeAg, hepatitis B e antigen; HBV DNA, hepatitis B virus DNA. ^#^ Units of measurement: ALT, IU/L; HBsAg, log_10_ IU/mL; HBeAg, log_10_ SCO; HBV DNA, log_10_ IU/mL.

## Data Availability

Data are available on request due to restrictions, e.g., privacy or ethical.

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
