# Peer review of "Quantitative HBsAg versus HBV DNA in Predicting Significant Hepatitis Activity of HBeAg-Positive Chronic HBV Infection"

_jcm, 2021, doi:10.3390/jcm10235617_

Round 1

Reviewer 1 Report

This study investigated the ALT plus HBV virologic parameters to correlate with liver pathologic activity. Authors utilized statistical methods to define high/low HBV replication and to examine the accuracy of ALT/HBV virologic parameters in predicting liver pathology injury. Authors concluded that Quantitative HBsAg instead of HBV 29 DNA is valuable in predicting significant hepatitis activity of HBeAg-positive CHB

Comments

  1. The manuscript is not reader friendly. Although the analysis or results are predictable, authors seem to describe or analyze the data in complex manner.
  2. Table 2-4 are not well described in text.
  3. A concern in the application of ALT is that the fluctuated features make the single measurement of ALT may compromise the analyzed results. Repeated measurement of ALT or analysis based on histology results, not including ALT, may be a more proper way.
  4. Prior studies revealed that a lower HBsAg is associated with advanced fibrosis in eAg+ CHB and also influenced by genotype of HBV. How about the impact of genotype on the study results?
  5. In figure 2, why only used ALT stratification in ROC curve analysis? It needs clarification.
  6. Baseline characteristics should be supplemented in Table 1, such as BW or BMI, comorbid diseases, or even steatosis grade in liver histology, etc. Some of them may have impact on the presentation of ALT level.

Author Response

Reviewer #1

This study investigated the ALT plus HBV virologic parameters to correlate with liver pathologic activity. Authors utilized statistical methods to define high/low HBV replication and to examine the accuracy of ALT/HBV virologic parameters in predicting liver pathology injury. Authors concluded that Quantitative HBsAg instead of HBV DNA is valuable in predicting significant hepatitis activity of HBeAg-positive CHB

Point 1: The manuscript is not reader friendly. Although the analysis or results are predictable, authors seem to describe or analyze the data in complex manner.

Response 1: Please see the penultimate paragraph of "Introduction" as well as the questions 5 to 7 of "Appendix" of the original manuscript.

  • Almost all previous studies refer to the existing natural history phase standards covering HBV DNA, which will lead to the actual importance of HBsAg underestimated because of the inherent correlation between HBV DNA and HBsAg. The importance of evaluating HBsAg should not be based on the natural history phase standards covering HBV DNA.
  • Almost all previous studies have HBeAg-positive patients as a whole, which will also lead to the actual importance of HBsAg underestimated because of the levels of HBV replication that affect HBV antigen expression are not stratified. In the "non-significant hepatitis activity" phase and the "significant hepatitis activity" phase of HBeAg-positive patients, the evolution direction of different HBV antigens and HBV DNA related to the levels of "hepatitis activity" in different stratifications is not exactly the same.
  • How to "reasonably" evaluate the importance of HBsAg and HBV DNA in predicting the natural history phases, and how to "reasonably" determine the cutoffs of HBsAg and HBV DNA in predicting the natural history phases, is the key problems that almost all previous studies have not solved, which is the main purpose of our manuscript.

Point 2: Table 2-4 are not well described in text.

Response 2: We may miss some content related to tables 2 to 4, which must be described in the text. However, we don't understand what not mentioned in the tables must be added to the text.

Point 3: A concern in the application of ALT is that the fluctuated features make the single measurement of ALT may compromise the analyzed results. Repeated measurement of ALT or analysis based on histology results, not including ALT, may be a more proper way.

Response 3: Please see the third paragraph of “Introduction”, table 4 of “Results” and the fifth paragraph of “Discussion”, as well as question 8 of “Appendix” of the original manuscript.

  • Liver pathological status is the best standard (not the gold standard) for determining “significant hepatitis activity”. However, a few patients with “significant hepatitis activity” always show mild liver necro-inflammation and slight liver fibrosis. In addition, there are sample errors as well as inter-observer and intra-observer bias in the evaluation of liver pathological status. Practically, repeated liver pathological examination is unrealistic.
  • ALT is a sensitive index to predict “significant hepatitis activity”. Theoretically, the abnormal increase of ALT levels indicates “significant hepatitis activity”. The fluctuation of ALT in patients with chronic HBV infection can be divided into fluctuation below the upper limit of normal reference and fluctuation above the upper limit of normal reference. Only the fluctuation above the upper limit of normal reference can indicate “significant hepatitis activity”, and only the fluctuation persistently below the upper limit of normal reference can indicate “non-significant hepatitis activity”. Practically, the upper limit of normal reference of ALT is still controversial, and the standard of upper limit of normal reference of ALT has not been unified, which has brought confusion to clinical practice and research. The results of previous studies and this study show that, even if ALT < 20 IU / L, about 10% of patients have extensive liver fibrosis (higher than or equal to S3) and about 5% of patients have liver cirrhosis (higher than or equal to S4). Therefore, it is inaccurate to predict “significant hepatitis activity” only based on the dynamic changes of ALT.
  • Accurate prediction of “significant hepatitis activity” also requires reference to other noninvasive parameters. Other noninvasive parameters that predict “significant hepatitis activity” in current clinical guidelines include at least HBV DNA and HBsAg. However, the rational cutoff for HBV DNA in predicting HBeAg-positive “significant hepatitis activity” has not been unified yet. Although EASL 2017 guidelines mentioned that HBsAg levels can be used as the basis for the phases of chronic HBV infection, but did not gave specific “quantization standard”.

Point 4: Prior studies revealed that a lower HBsAg is associated with advanced fibrosis in eAg+ CHB and also influenced by genotype of HBV. How about the impact of genotype on the study results?

Response 4: This is an issue that may have clinical importance. Unfortunately, this study is a retrospective cross-sectional study. Most patients lack HBV genotype data, which may be a flaw of this study. We look forward to future studies that have a similar design to this study, including HBV genotypes into analyses, to obtain more refined research results.

  • Previous studies based on the overall HBeAg-positive patients have reached a consistent conclusion: lower HBsAg has some predictive value for advanced fibrosis or cirrhosis, but whether advanced fibrosis or cirrhosis is more related to HBV genotype remains uncertain.
  • With the current availability of antiviral drugs, it may be more reasonable to manage patients based on the natural history phases than based on the stages of fibrosis. Therefore, exploring and evaluating the effective parameters that delimit the phases of natural history may be more clinically important.
  • Previous studies based on the natural history phase criteria covering HBVDNA and based on the overall HBeAg-positive patients showed that, HBsAg quantification "may be" related to HBV genotype. In other words, the correlation between HBsAg quantification and HBV genotype has not yet been determined.
  • Most of the subjects of this study are from East China, and the HBV genotypes of most of the subjects are genotype B and genotype C.

Point 5: In figure 2, why only used ALT stratification in ROC curve analysis? It needs clarification.

Response 5: Please see the question 8 of "Appendix" of the original manuscript.

  • The patients in HBeAg-positive "non-significant hepatitis activity" phase should be a highly homogenized population, whose characteristic must be sustained and stable "mild hepatic necro-inflammation, and no or slight hepatic fibrosis, and normal ALT". Therefore, the demarcation standard of HBeAg-positive "significant hepatitis activity" phase should be that "pathological grade higher than G1, or the pathological stage higher than S1, or ALT higher than or equal to ULNs".
  • On the contrary, the patients in HBeAg-positive "significant hepatitis activity" phase are a highly heterogenized population, whose behavior may be persistent or recurrent "significant hepatitis activity". In patients with early “significant hepatitis activity”, the pathological grade of higher than G1 or the stage of higher than S1 does not necessarily occur; in patients with late “significant hepatitis activity”, the high levels of pathological grade and stage may return to G1 and S1. In addition, in a few patients with “significant hepatitis activity”, the pathological grade may be never higher than G1 and the stage may be never higher than S1. Therefore, in the population separated from the "long term stable non significant hepatitis activity", i.e. the population entered “significant hepatitis activity” phase, further identifying the subpopulation who are still active, should only refer to the standards of " ALT higher than or equal to ULNs".

Point 6: Baseline characteristics should be supplemented in Table 1, such as BW or BMI, comorbid diseases, or even steatosis grade in liver histology, etc. Some of them may have impact on the presentation of ALT level.

Response 6: Please see the description of the “Study population” of “Materials and Methods” of the original manuscript.

Some comorbid diseases, including coinfection with other hepatotropic viruses, Epstein-Barr virus, cytomegalovirus; Schistosomiasis japonica liver disease, nonalcoholic/metabolic fatty liver disease, drug-induced liver injury, excessive drinking, autoimmune diseases, thyroid diseases, and gallstones and bile duct stones, have been excluded.

Reviewer #1

This study investigated the ALT plus HBV virologic parameters to correlate with liver pathologic activity. Authors utilized statistical methods to define high/low HBV replication and to examine the accuracy of ALT/HBV virologic parameters in predicting liver pathology injury. Authors concluded that Quantitative HBsAg instead of HBV DNA is valuable in predicting significant hepatitis activity of HBeAg-positive CHB

Point 1: The manuscript is not reader friendly. Although the analysis or results are predictable, authors seem to describe or analyze the data in complex manner.

Response 1: Please see the penultimate paragraph of "Introduction" as well as the questions 5 to 7 of "Appendix" of the original manuscript.

  • Almost all previous studies refer to the existing natural history phase standards covering HBV DNA, which will lead to the actual importance of HBsAg underestimated because of the inherent correlation between HBV DNA and HBsAg. The importance of evaluating HBsAg should not be based on the natural history phase standards covering HBV DNA.
  • Almost all previous studies have HBeAg-positive patients as a whole, which will also lead to the actual importance of HBsAg underestimated because of the levels of HBV replication that affect HBV antigen expression are not stratified. In the "non-significant hepatitis activity" phase and the "significant hepatitis activity" phase of HBeAg-positive patients, the evolution direction of different HBV antigens and HBV DNA related to the levels of "hepatitis activity" in different stratifications is not exactly the same.
  • How to "reasonably" evaluate the importance of HBsAg and HBV DNA in predicting the natural history phases, and how to "reasonably" determine the cutoffs of HBsAg and HBV DNA in predicting the natural history phases, is the key problems that almost all previous studies have not solved, which is the main purpose of our manuscript.

Point 2: Table 2-4 are not well described in text.

Response 2: We may miss some content related to tables 2 to 4, which must be described in the text. However, we don't understand what not mentioned in the tables must be added to the text.

Point 3: A concern in the application of ALT is that the fluctuated features make the single measurement of ALT may compromise the analyzed results. Repeated measurement of ALT or analysis based on histology results, not including ALT, may be a more proper way.

Response 3: Please see the third paragraph of “Introduction”, table 4 of “Results” and the fifth paragraph of “Discussion”, as well as question 8 of “Appendix” of the original manuscript.

  • Liver pathological status is the best standard (not the gold standard) for determining “significant hepatitis activity”. However, a few patients with “significant hepatitis activity” always show mild liver necro-inflammation and slight liver fibrosis. In addition, there are sample errors as well as inter-observer and intra-observer bias in the evaluation of liver pathological status. Practically, repeated liver pathological examination is unrealistic.
  • ALT is a sensitive index to predict “significant hepatitis activity”. Theoretically, the abnormal increase of ALT levels indicates “significant hepatitis activity”. The fluctuation of ALT in patients with chronic HBV infection can be divided into fluctuation below the upper limit of normal reference and fluctuation above the upper limit of normal reference. Only the fluctuation above the upper limit of normal reference can indicate “significant hepatitis activity”, and only the fluctuation persistently below the upper limit of normal reference can indicate “non-significant hepatitis activity”. Practically, the upper limit of normal reference of ALT is still controversial, and the standard of upper limit of normal reference of ALT has not been unified, which has brought confusion to clinical practice and research. The results of previous studies and this study show that, even if ALT < 20 IU / L, about 10% of patients have extensive liver fibrosis (higher than or equal to S3) and about 5% of patients have liver cirrhosis (higher than or equal to S4). Therefore, it is inaccurate to predict “significant hepatitis activity” only based on the dynamic changes of ALT.
  • Accurate prediction of “significant hepatitis activity” also requires reference to other noninvasive parameters. Other noninvasive parameters that predict “significant hepatitis activity” in current clinical guidelines include at least HBV DNA and HBsAg. However, the rational cutoff for HBV DNA in predicting HBeAg-positive “significant hepatitis activity” has not been unified yet. Although EASL 2017 guidelines mentioned that HBsAg levels can be used as the basis for the phases of chronic HBV infection, but did not gave specific “quantization standard”.

Point 4: Prior studies revealed that a lower HBsAg is associated with advanced fibrosis in eAg+ CHB and also influenced by genotype of HBV. How about the impact of genotype on the study results?

Response 4: This is an issue that may have clinical importance. Unfortunately, this study is a retrospective cross-sectional study. Most patients lack HBV genotype data, which may be a flaw of this study. We look forward to future studies that have a similar design to this study, including HBV genotypes into analyses, to obtain more refined research results.

  • Previous studies based on the overall HBeAg-positive patients have reached a consistent conclusion: lower HBsAg has some predictive value for advanced fibrosis or cirrhosis, but whether advanced fibrosis or cirrhosis is more related to HBV genotype remains uncertain.
  • With the current availability of antiviral drugs, it may be more reasonable to manage patients based on the natural history phases than based on the stages of fibrosis. Therefore, exploring and evaluating the effective parameters that delimit the phases of natural history may be more clinically important.
  • Previous studies based on the natural history phase criteria covering HBVDNA and based on the overall HBeAg-positive patients showed that, HBsAg quantification "may be" related to HBV genotype. In other words, the correlation between HBsAg quantification and HBV genotype has not yet been determined.
  • Most of the subjects of this study are from East China, and the HBV genotypes of most of the subjects are genotype B and genotype C.

Point 5: In figure 2, why only used ALT stratification in ROC curve analysis? It needs clarification.

Response 5: Please see the question 8 of "Appendix" of the original manuscript.

  • The patients in HBeAg-positive "non-significant hepatitis activity" phase should be a highly homogenized population, whose characteristic must be sustained and stable "mild hepatic necro-inflammation, and no or slight hepatic fibrosis, and normal ALT". Therefore, the demarcation standard of HBeAg-positive "significant hepatitis activity" phase should be that "pathological grade higher than G1, or the pathological stage higher than S1, or ALT higher than or equal to ULNs".
  • On the contrary, the patients in HBeAg-positive "significant hepatitis activity" phase are a highly heterogenized population, whose behavior may be persistent or recurrent "significant hepatitis activity". In patients with early “significant hepatitis activity”, the pathological grade of higher than G1 or the stage of higher than S1 does not necessarily occur; in patients with late “significant hepatitis activity”, the high levels of pathological grade and stage may return to G1 and S1. In addition, in a few patients with “significant hepatitis activity”, the pathological grade may be never higher than G1 and the stage may be never higher than S1. Therefore, in the population separated from the "long term stable non significant hepatitis activity", i.e. the population entered “significant hepatitis activity” phase, further identifying the subpopulation who are still active, should only refer to the standards of " ALT higher than or equal to ULNs".

Point 6: Baseline characteristics should be supplemented in Table 1, such as BW or BMI, comorbid diseases, or even steatosis grade in liver histology, etc. Some of them may have impact on the presentation of ALT level.

Response 6: Please see the description of the “Study population” of “Materials and Methods” of the original manuscript.

Some comorbid diseases, including coinfection with other hepatotropic viruses, Epstein-Barr virus, cytomegalovirus; Schistosomiasis japonica liver disease, nonalcoholic/metabolic fatty liver disease, drug-induced liver injury, excessive drinking, autoimmune diseases, thyroid diseases, and gallstones and bile duct stones, have been excluded.

Reviewer #1

This study investigated the ALT plus HBV virologic parameters to correlate with liver pathologic activity. Authors utilized statistical methods to define high/low HBV replication and to examine the accuracy of ALT/HBV virologic parameters in predicting liver pathology injury. Authors concluded that Quantitative HBsAg instead of HBV DNA is valuable in predicting significant hepatitis activity of HBeAg-positive CHB

Point 1: The manuscript is not reader friendly. Although the analysis or results are predictable, authors seem to describe or analyze the data in complex manner.

Response 1: Please see the penultimate paragraph of "Introduction" as well as the questions 5 to 7 of "Appendix" of the original manuscript.

  • Almost all previous studies refer to the existing natural history phase standards covering HBV DNA, which will lead to the actual importance of HBsAg underestimated because of the inherent correlation between HBV DNA and HBsAg. The importance of evaluating HBsAg should not be based on the natural history phase standards covering HBV DNA.
  • Almost all previous studies have HBeAg-positive patients as a whole, which will also lead to the actual importance of HBsAg underestimated because of the levels of HBV replication that affect HBV antigen expression are not stratified. In the "non-significant hepatitis activity" phase and the "significant hepatitis activity" phase of HBeAg-positive patients, the evolution direction of different HBV antigens and HBV DNA related to the levels of "hepatitis activity" in different stratifications is not exactly the same.
  • How to "reasonably" evaluate the importance of HBsAg and HBV DNA in predicting the natural history phases, and how to "reasonably" determine the cutoffs of HBsAg and HBV DNA in predicting the natural history phases, is the key problems that almost all previous studies have not solved, which is the main purpose of our manuscript.

Point 2: Table 2-4 are not well described in text.

Response 2: We may miss some content related to tables 2 to 4, which must be described in the text. However, we don't understand what not mentioned in the tables must be added to the text.

Point 3: A concern in the application of ALT is that the fluctuated features make the single measurement of ALT may compromise the analyzed results. Repeated measurement of ALT or analysis based on histology results, not including ALT, may be a more proper way.

Response 3: Please see the third paragraph of “Introduction”, table 4 of “Results” and the fifth paragraph of “Discussion”, as well as question 8 of “Appendix” of the original manuscript.

  • Liver pathological status is the best standard (not the gold standard) for determining “significant hepatitis activity”. However, a few patients with “significant hepatitis activity” always show mild liver necro-inflammation and slight liver fibrosis. In addition, there are sample errors as well as inter-observer and intra-observer bias in the evaluation of liver pathological status. Practically, repeated liver pathological examination is unrealistic.
  • ALT is a sensitive index to predict “significant hepatitis activity”. Theoretically, the abnormal increase of ALT levels indicates “significant hepatitis activity”. The fluctuation of ALT in patients with chronic HBV infection can be divided into fluctuation below the upper limit of normal reference and fluctuation above the upper limit of normal reference. Only the fluctuation above the upper limit of normal reference can indicate “significant hepatitis activity”, and only the fluctuation persistently below the upper limit of normal reference can indicate “non-significant hepatitis activity”. Practically, the upper limit of normal reference of ALT is still controversial, and the standard of upper limit of normal reference of ALT has not been unified, which has brought confusion to clinical practice and research. The results of previous studies and this study show that, even if ALT < 20 IU / L, about 10% of patients have extensive liver fibrosis (higher than or equal to S3) and about 5% of patients have liver cirrhosis (higher than or equal to S4). Therefore, it is inaccurate to predict “significant hepatitis activity” only based on the dynamic changes of ALT.
  • Accurate prediction of “significant hepatitis activity” also requires reference to other noninvasive parameters. Other noninvasive parameters that predict “significant hepatitis activity” in current clinical guidelines include at least HBV DNA and HBsAg. However, the rational cutoff for HBV DNA in predicting HBeAg-positive “significant hepatitis activity” has not been unified yet. Although EASL 2017 guidelines mentioned that HBsAg levels can be used as the basis for the phases of chronic HBV infection, but did not gave specific “quantization standard”.

Point 4: Prior studies revealed that a lower HBsAg is associated with advanced fibrosis in eAg+ CHB and also influenced by genotype of HBV. How about the impact of genotype on the study results?

Response 4: This is an issue that may have clinical importance. Unfortunately, this study is a retrospective cross-sectional study. Most patients lack HBV genotype data, which may be a flaw of this study. We look forward to future studies that have a similar design to this study, including HBV genotypes into analyses, to obtain more refined research results.

  • Previous studies based on the overall HBeAg-positive patients have reached a consistent conclusion: lower HBsAg has some predictive value for advanced fibrosis or cirrhosis, but whether advanced fibrosis or cirrhosis is more related to HBV genotype remains uncertain.
  • With the current availability of antiviral drugs, it may be more reasonable to manage patients based on the natural history phases than based on the stages of fibrosis. Therefore, exploring and evaluating the effective parameters that delimit the phases of natural history may be more clinically important.
  • Previous studies based on the natural history phase criteria covering HBVDNA and based on the overall HBeAg-positive patients showed that, HBsAg quantification "may be" related to HBV genotype. In other words, the correlation between HBsAg quantification and HBV genotype has not yet been determined.
  • Most of the subjects of this study are from East China, and the HBV genotypes of most of the subjects are genotype B and genotype C.

Point 5: In figure 2, why only used ALT stratification in ROC curve analysis? It needs clarification.

Response 5: Please see the question 8 of "Appendix" of the original manuscript.

  • The patients in HBeAg-positive "non-significant hepatitis activity" phase should be a highly homogenized population, whose characteristic must be sustained and stable "mild hepatic necro-inflammation, and no or slight hepatic fibrosis, and normal ALT". Therefore, the demarcation standard of HBeAg-positive "significant hepatitis activity" phase should be that "pathological grade higher than G1, or the pathological stage higher than S1, or ALT higher than or equal to ULNs".
  • On the contrary, the patients in HBeAg-positive "significant hepatitis activity" phase are a highly heterogenized population, whose behavior may be persistent or recurrent "significant hepatitis activity". In patients with early “significant hepatitis activity”, the pathological grade of higher than G1 or the stage of higher than S1 does not necessarily occur; in patients with late “significant hepatitis activity”, the high levels of pathological grade and stage may return to G1 and S1. In addition, in a few patients with “significant hepatitis activity”, the pathological grade may be never higher than G1 and the stage may be never higher than S1. Therefore, in the population separated from the "long term stable non significant hepatitis activity", i.e. the population entered “significant hepatitis activity” phase, further identifying the subpopulation who are still active, should only refer to the standards of " ALT higher than or equal to ULNs".

Point 6: Baseline characteristics should be supplemented in Table 1, such as BW or BMI, comorbid diseases, or even steatosis grade in liver histology, etc. Some of them may have impact on the presentation of ALT level.

Response 6: Please see the description of the “Study population” of “Materials and Methods” of the original manuscript.

Some comorbid diseases, including coinfection with other hepatotropic viruses, Epstein-Barr virus, cytomegalovirus; Schistosomiasis japonica liver disease, nonalcoholic/metabolic fatty liver disease, drug-induced liver injury, excessive drinking, autoimmune diseases, thyroid diseases, and gallstones and bile duct stones, have been excluded.

Reviewer #1

This study investigated the ALT plus HBV virologic parameters to correlate with liver pathologic activity. Authors utilized statistical methods to define high/low HBV replication and to examine the accuracy of ALT/HBV virologic parameters in predicting liver pathology injury. Authors concluded that Quantitative HBsAg instead of HBV DNA is valuable in predicting significant hepatitis activity of HBeAg-positive CHB

Point 1: The manuscript is not reader friendly. Although the analysis or results are predictable, authors seem to describe or analyze the data in complex manner.

Response 1: Please see the penultimate paragraph of "Introduction" as well as the questions 5 to 7 of "Appendix" of the original manuscript.

  • Almost all previous studies refer to the existing natural history phase standards covering HBV DNA, which will lead to the actual importance of HBsAg underestimated because of the inherent correlation between HBV DNA and HBsAg. The importance of evaluating HBsAg should not be based on the natural history phase standards covering HBV DNA.
  • Almost all previous studies have HBeAg-positive patients as a whole, which will also lead to the actual importance of HBsAg underestimated because of the levels of HBV replication that affect HBV antigen expression are not stratified. In the "non-significant hepatitis activity" phase and the "significant hepatitis activity" phase of HBeAg-positive patients, the evolution direction of different HBV antigens and HBV DNA related to the levels of "hepatitis activity" in different stratifications is not exactly the same.
  • How to "reasonably" evaluate the importance of HBsAg and HBV DNA in predicting the natural history phases, and how to "reasonably" determine the cutoffs of HBsAg and HBV DNA in predicting the natural history phases, is the key problems that almost all previous studies have not solved, which is the main purpose of our manuscript.

Point 2: Table 2-4 are not well described in text.

Response 2: We may miss some content related to tables 2 to 4, which must be described in the text. However, we don't understand what not mentioned in the tables must be added to the text.

Point 3: A concern in the application of ALT is that the fluctuated features make the single measurement of ALT may compromise the analyzed results. Repeated measurement of ALT or analysis based on histology results, not including ALT, may be a more proper way.

Response 3: Please see the third paragraph of “Introduction”, table 4 of “Results” and the fifth paragraph of “Discussion”, as well as question 8 of “Appendix” of the original manuscript.

  • Liver pathological status is the best standard (not the gold standard) for determining “significant hepatitis activity”. However, a few patients with “significant hepatitis activity” always show mild liver necro-inflammation and slight liver fibrosis. In addition, there are sample errors as well as inter-observer and intra-observer bias in the evaluation of liver pathological status. Practically, repeated liver pathological examination is unrealistic.
  • ALT is a sensitive index to predict “significant hepatitis activity”. Theoretically, the abnormal increase of ALT levels indicates “significant hepatitis activity”. The fluctuation of ALT in patients with chronic HBV infection can be divided into fluctuation below the upper limit of normal reference and fluctuation above the upper limit of normal reference. Only the fluctuation above the upper limit of normal reference can indicate “significant hepatitis activity”, and only the fluctuation persistently below the upper limit of normal reference can indicate “non-significant hepatitis activity”. Practically, the upper limit of normal reference of ALT is still controversial, and the standard of upper limit of normal reference of ALT has not been unified, which has brought confusion to clinical practice and research. The results of previous studies and this study show that, even if ALT < 20 IU / L, about 10% of patients have extensive liver fibrosis (higher than or equal to S3) and about 5% of patients have liver cirrhosis (higher than or equal to S4). Therefore, it is inaccurate to predict “significant hepatitis activity” only based on the dynamic changes of ALT.
  • Accurate prediction of “significant hepatitis activity” also requires reference to other noninvasive parameters. Other noninvasive parameters that predict “significant hepatitis activity” in current clinical guidelines include at least HBV DNA and HBsAg. However, the rational cutoff for HBV DNA in predicting HBeAg-positive “significant hepatitis activity” has not been unified yet. Although EASL 2017 guidelines mentioned that HBsAg levels can be used as the basis for the phases of chronic HBV infection, but did not gave specific “quantization standard”.

Point 4: Prior studies revealed that a lower HBsAg is associated with advanced fibrosis in eAg+ CHB and also influenced by genotype of HBV. How about the impact of genotype on the study results?

Response 4: This is an issue that may have clinical importance. Unfortunately, this study is a retrospective cross-sectional study. Most patients lack HBV genotype data, which may be a flaw of this study. We look forward to future studies that have a similar design to this study, including HBV genotypes into analyses, to obtain more refined research results.

  • Previous studies based on the overall HBeAg-positive patients have reached a consistent conclusion: lower HBsAg has some predictive value for advanced fibrosis or cirrhosis, but whether advanced fibrosis or cirrhosis is more related to HBV genotype remains uncertain.
  • With the current availability of antiviral drugs, it may be more reasonable to manage patients based on the natural history phases than based on the stages of fibrosis. Therefore, exploring and evaluating the effective parameters that delimit the phases of natural history may be more clinically important.
  • Previous studies based on the natural history phase criteria covering HBVDNA and based on the overall HBeAg-positive patients showed that, HBsAg quantification "may be" related to HBV genotype. In other words, the correlation between HBsAg quantification and HBV genotype has not yet been determined.
  • Most of the subjects of this study are from East China, and the HBV genotypes of most of the subjects are genotype B and genotype C.

Point 5: In figure 2, why only used ALT stratification in ROC curve analysis? It needs clarification.

Response 5: Please see the question 8 of "Appendix" of the original manuscript.

  • The patients in HBeAg-positive "non-significant hepatitis activity" phase should be a highly homogenized population, whose characteristic must be sustained and stable "mild hepatic necro-inflammation, and no or slight hepatic fibrosis, and normal ALT". Therefore, the demarcation standard of HBeAg-positive "significant hepatitis activity" phase should be that "pathological grade higher than G1, or the pathological stage higher than S1, or ALT higher than or equal to ULNs".
  • On the contrary, the patients in HBeAg-positive "significant hepatitis activity" phase are a highly heterogenized population, whose behavior may be persistent or recurrent "significant hepatitis activity". In patients with early “significant hepatitis activity”, the pathological grade of higher than G1 or the stage of higher than S1 does not necessarily occur; in patients with late “significant hepatitis activity”, the high levels of pathological grade and stage may return to G1 and S1. In addition, in a few patients with “significant hepatitis activity”, the pathological grade may be never higher than G1 and the stage may be never higher than S1. Therefore, in the population separated from the "long term stable non significant hepatitis activity", i.e. the population entered “significant hepatitis activity” phase, further identifying the subpopulation who are still active, should only refer to the standards of " ALT higher than or equal to ULNs".

Point 6: Baseline characteristics should be supplemented in Table 1, such as BW or BMI, comorbid diseases, or even steatosis grade in liver histology, etc. Some of them may have impact on the presentation of ALT level.

Response 6: Please see the description of the “Study population” of “Materials and Methods” of the original manuscript.

Some comorbid diseases, including coinfection with other hepatotropic viruses, Epstein-Barr virus, cytomegalovirus; Schistosomiasis japonica liver disease, nonalcoholic/metabolic fatty liver disease, drug-induced liver injury, excessive drinking, autoimmune diseases, thyroid diseases, and gallstones and bile duct stones, have been excluded.

Reviewer #1

This study investigated the ALT plus HBV virologic parameters to correlate with liver pathologic activity. Authors utilized statistical methods to define high/low HBV replication and to examine the accuracy of ALT/HBV virologic parameters in predicting liver pathology injury. Authors concluded that Quantitative HBsAg instead of HBV DNA is valuable in predicting significant hepatitis activity of HBeAg-positive CHB

Point 1: The manuscript is not reader friendly. Although the analysis or results are predictable, authors seem to describe or analyze the data in complex manner.

Response 1: Please see the penultimate paragraph of "Introduction" as well as the questions 5 to 7 of "Appendix" of the original manuscript.

  • Almost all previous studies refer to the existing natural history phase standards covering HBV DNA, which will lead to the actual importance of HBsAg underestimated because of the inherent correlation between HBV DNA and HBsAg. The importance of evaluating HBsAg should not be based on the natural history phase standards covering HBV DNA.
  • Almost all previous studies have HBeAg-positive patients as a whole, which will also lead to the actual importance of HBsAg underestimated because of the levels of HBV replication that affect HBV antigen expression are not stratified. In the "non-significant hepatitis activity" phase and the "significant hepatitis activity" phase of HBeAg-positive patients, the evolution direction of different HBV antigens and HBV DNA related to the levels of "hepatitis activity" in different stratifications is not exactly the same.
  • How to "reasonably" evaluate the importance of HBsAg and HBV DNA in predicting the natural history phases, and how to "reasonably" determine the cutoffs of HBsAg and HBV DNA in predicting the natural history phases, is the key problems that almost all previous studies have not solved, which is the main purpose of our manuscript.

Point 2: Table 2-4 are not well described in text.

Response 2: We may miss some content related to tables 2 to 4, which must be described in the text. However, we don't understand what not mentioned in the tables must be added to the text.

Point 3: A concern in the application of ALT is that the fluctuated features make the single measurement of ALT may compromise the analyzed results. Repeated measurement of ALT or analysis based on histology results, not including ALT, may be a more proper way.

Response 3: Please see the third paragraph of “Introduction”, table 4 of “Results” and the fifth paragraph of “Discussion”, as well as question 8 of “Appendix” of the original manuscript.

  • Liver pathological status is the best standard (not the gold standard) for determining “significant hepatitis activity”. However, a few patients with “significant hepatitis activity” always show mild liver necro-inflammation and slight liver fibrosis. In addition, there are sample errors as well as inter-observer and intra-observer bias in the evaluation of liver pathological status. Practically, repeated liver pathological examination is unrealistic.
  • ALT is a sensitive index to predict “significant hepatitis activity”. Theoretically, the abnormal increase of ALT levels indicates “significant hepatitis activity”. The fluctuation of ALT in patients with chronic HBV infection can be divided into fluctuation below the upper limit of normal reference and fluctuation above the upper limit of normal reference. Only the fluctuation above the upper limit of normal reference can indicate “significant hepatitis activity”, and only the fluctuation persistently below the upper limit of normal reference can indicate “non-significant hepatitis activity”. Practically, the upper limit of normal reference of ALT is still controversial, and the standard of upper limit of normal reference of ALT has not been unified, which has brought confusion to clinical practice and research. The results of previous studies and this study show that, even if ALT < 20 IU / L, about 10% of patients have extensive liver fibrosis (higher than or equal to S3) and about 5% of patients have liver cirrhosis (higher than or equal to S4). Therefore, it is inaccurate to predict “significant hepatitis activity” only based on the dynamic changes of ALT.
  • Accurate prediction of “significant hepatitis activity” also requires reference to other noninvasive parameters. Other noninvasive parameters that predict “significant hepatitis activity” in current clinical guidelines include at least HBV DNA and HBsAg. However, the rational cutoff for HBV DNA in predicting HBeAg-positive “significant hepatitis activity” has not been unified yet. Although EASL 2017 guidelines mentioned that HBsAg levels can be used as the basis for the phases of chronic HBV infection, but did not gave specific “quantization standard”.

Point 4: Prior studies revealed that a lower HBsAg is associated with advanced fibrosis in eAg+ CHB and also influenced by genotype of HBV. How about the impact of genotype on the study results?

Response 4: This is an issue that may have clinical importance. Unfortunately, this study is a retrospective cross-sectional study. Most patients lack HBV genotype data, which may be a flaw of this study. We look forward to future studies that have a similar design to this study, including HBV genotypes into analyses, to obtain more refined research results.

  • Previous studies based on the overall HBeAg-positive patients have reached a consistent conclusion: lower HBsAg has some predictive value for advanced fibrosis or cirrhosis, but whether advanced fibrosis or cirrhosis is more related to HBV genotype remains uncertain.
  • With the current availability of antiviral drugs, it may be more reasonable to manage patients based on the natural history phases than based on the stages of fibrosis. Therefore, exploring and evaluating the effective parameters that delimit the phases of natural history may be more clinically important.
  • Previous studies based on the natural history phase criteria covering HBVDNA and based on the overall HBeAg-positive patients showed that, HBsAg quantification "may be" related to HBV genotype. In other words, the correlation between HBsAg quantification and HBV genotype has not yet been determined.
  • Most of the subjects of this study are from East China, and the HBV genotypes of most of the subjects are genotype B and genotype C.

Point 5: In figure 2, why only used ALT stratification in ROC curve analysis? It needs clarification.

Response 5: Please see the question 8 of "Appendix" of the original manuscript.

  • The patients in HBeAg-positive "non-significant hepatitis activity" phase should be a highly homogenized population, whose characteristic must be sustained and stable "mild hepatic necro-inflammation, and no or slight hepatic fibrosis, and normal ALT". Therefore, the demarcation standard of HBeAg-positive "significant hepatitis activity" phase should be that "pathological grade higher than G1, or the pathological stage higher than S1, or ALT higher than or equal to ULNs".
  • On the contrary, the patients in HBeAg-positive "significant hepatitis activity" phase are a highly heterogenized population, whose behavior may be persistent or recurrent "significant hepatitis activity". In patients with early “significant hepatitis activity”, the pathological grade of higher than G1 or the stage of higher than S1 does not necessarily occur; in patients with late “significant hepatitis activity”, the high levels of pathological grade and stage may return to G1 and S1. In addition, in a few patients with “significant hepatitis activity”, the pathological grade may be never higher than G1 and the stage may be never higher than S1. Therefore, in the population separated from the "long term stable non significant hepatitis activity", i.e. the population entered “significant hepatitis activity” phase, further identifying the subpopulation who are still active, should only refer to the standards of " ALT higher than or equal to ULNs".

Point 6: Baseline characteristics should be supplemented in Table 1, such as BW or BMI, comorbid diseases, or even steatosis grade in liver histology, etc. Some of them may have impact on the presentation of ALT level.

Response 6: Please see the description of the “Study population” of “Materials and Methods” of the original manuscript.

Some comorbid diseases, including coinfection with other hepatotropic viruses, Epstein-Barr virus, cytomegalovirus; Schistosomiasis japonica liver disease, nonalcoholic/metabolic fatty liver disease, drug-induced liver injury, excessive drinking, autoimmune diseases, thyroid diseases, and gallstones and bile duct stones, have been excluded.

Reviewer #1

This study investigated the ALT plus HBV virologic parameters to correlate with liver pathologic activity. Authors utilized statistical methods to define high/low HBV replication and to examine the accuracy of ALT/HBV virologic parameters in predicting liver pathology injury. Authors concluded that Quantitative HBsAg instead of HBV DNA is valuable in predicting significant hepatitis activity of HBeAg-positive CHB

Point 1: The manuscript is not reader friendly. Although the analysis or results are predictable, authors seem to describe or analyze the data in complex manner.

Response 1: Please see the penultimate paragraph of "Introduction" as well as the questions 5 to 7 of "Appendix" of the original manuscript.

  • Almost all previous studies refer to the existing natural history phase standards covering HBV DNA, which will lead to the actual importance of HBsAg underestimated because of the inherent correlation between HBV DNA and HBsAg. The importance of evaluating HBsAg should not be based on the natural history phase standards covering HBV DNA.
  • Almost all previous studies have HBeAg-positive patients as a whole, which will also lead to the actual importance of HBsAg underestimated because of the levels of HBV replication that affect HBV antigen expression are not stratified. In the "non-significant hepatitis activity" phase and the "significant hepatitis activity" phase of HBeAg-positive patients, the evolution direction of different HBV antigens and HBV DNA related to the levels of "hepatitis activity" in different stratifications is not exactly the same.
  • How to "reasonably" evaluate the importance of HBsAg and HBV DNA in predicting the natural history phases, and how to "reasonably" determine the cutoffs of HBsAg and HBV DNA in predicting the natural history phases, is the key problems that almost all previous studies have not solved, which is the main purpose of our manuscript.

Point 2: Table 2-4 are not well described in text.

Response 2: We may miss some content related to tables 2 to 4, which must be described in the text. However, we don't understand what not mentioned in the tables must be added to the text.

Point 3: A concern in the application of ALT is that the fluctuated features make the single measurement of ALT may compromise the analyzed results. Repeated measurement of ALT or analysis based on histology results, not including ALT, may be a more proper way.

Response 3: Please see the third paragraph of “Introduction”, table 4 of “Results” and the fifth paragraph of “Discussion”, as well as question 8 of “Appendix” of the original manuscript.

  • Liver pathological status is the best standard (not the gold standard) for determining “significant hepatitis activity”. However, a few patients with “significant hepatitis activity” always show mild liver necro-inflammation and slight liver fibrosis. In addition, there are sample errors as well as inter-observer and intra-observer bias in the evaluation of liver pathological status. Practically, repeated liver pathological examination is unrealistic.
  • ALT is a sensitive index to predict “significant hepatitis activity”. Theoretically, the abnormal increase of ALT levels indicates “significant hepatitis activity”. The fluctuation of ALT in patients with chronic HBV infection can be divided into fluctuation below the upper limit of normal reference and fluctuation above the upper limit of normal reference. Only the fluctuation above the upper limit of normal reference can indicate “significant hepatitis activity”, and only the fluctuation persistently below the upper limit of normal reference can indicate “non-significant hepatitis activity”. Practically, the upper limit of normal reference of ALT is still controversial, and the standard of upper limit of normal reference of ALT has not been unified, which has brought confusion to clinical practice and research. The results of previous studies and this study show that, even if ALT < 20 IU / L, about 10% of patients have extensive liver fibrosis (higher than or equal to S3) and about 5% of patients have liver cirrhosis (higher than or equal to S4). Therefore, it is inaccurate to predict “significant hepatitis activity” only based on the dynamic changes of ALT.
  • Accurate prediction of “significant hepatitis activity” also requires reference to other noninvasive parameters. Other noninvasive parameters that predict “significant hepatitis activity” in current clinical guidelines include at least HBV DNA and HBsAg. However, the rational cutoff for HBV DNA in predicting HBeAg-positive “significant hepatitis activity” has not been unified yet. Although EASL 2017 guidelines mentioned that HBsAg levels can be used as the basis for the phases of chronic HBV infection, but did not gave specific “quantization standard”.

Point 4: Prior studies revealed that a lower HBsAg is associated with advanced fibrosis in eAg+ CHB and also influenced by genotype of HBV. How about the impact of genotype on the study results?

Response 4: This is an issue that may have clinical importance. Unfortunately, this study is a retrospective cross-sectional study. Most patients lack HBV genotype data, which may be a flaw of this study. We look forward to future studies that have a similar design to this study, including HBV genotypes into analyses, to obtain more refined research results.

  • Previous studies based on the overall HBeAg-positive patients have reached a consistent conclusion: lower HBsAg has some predictive value for advanced fibrosis or cirrhosis, but whether advanced fibrosis or cirrhosis is more related to HBV genotype remains uncertain.
  • With the current availability of antiviral drugs, it may be more reasonable to manage patients based on the natural history phases than based on the stages of fibrosis. Therefore, exploring and evaluating the effective parameters that delimit the phases of natural history may be more clinically important.
  • Previous studies based on the natural history phase criteria covering HBVDNA and based on the overall HBeAg-positive patients showed that, HBsAg quantification "may be" related to HBV genotype. In other words, the correlation between HBsAg quantification and HBV genotype has not yet been determined.
  • Most of the subjects of this study are from East China, and the HBV genotypes of most of the subjects are genotype B and genotype C.

Point 5: In figure 2, why only used ALT stratification in ROC curve analysis? It needs clarification.

Response 5: Please see the question 8 of "Appendix" of the original manuscript.

  • The patients in HBeAg-positive "non-significant hepatitis activity" phase should be a highly homogenized population, whose characteristic must be sustained and stable "mild hepatic necro-inflammation, and no or slight hepatic fibrosis, and normal ALT". Therefore, the demarcation standard of HBeAg-positive "significant hepatitis activity" phase should be that "pathological grade higher than G1, or the pathological stage higher than S1, or ALT higher than or equal to ULNs".
  • On the contrary, the patients in HBeAg-positive "significant hepatitis activity" phase are a highly heterogenized population, whose behavior may be persistent or recurrent "significant hepatitis activity". In patients with early “significant hepatitis activity”, the pathological grade of higher than G1 or the stage of higher than S1 does not necessarily occur; in patients with late “significant hepatitis activity”, the high levels of pathological grade and stage may return to G1 and S1. In addition, in a few patients with “significant hepatitis activity”, the pathological grade may be never higher than G1 and the stage may be never higher than S1. Therefore, in the population separated from the "long term stable non significant hepatitis activity", i.e. the population entered “significant hepatitis activity” phase, further identifying the subpopulation who are still active, should only refer to the standards of " ALT higher than or equal to ULNs".

Point 6: Baseline characteristics should be supplemented in Table 1, such as BW or BMI, comorbid diseases, or even steatosis grade in liver histology, etc. Some of them may have impact on the presentation of ALT level.

Response 6: Please see the description of the “Study population” of “Materials and Methods” of the original manuscript.

Some comorbid diseases, including coinfection with other hepatotropic viruses, Epstein-Barr virus, cytomegalovirus; Schistosomiasis japonica liver disease, nonalcoholic/metabolic fatty liver disease, drug-induced liver injury, excessive drinking, autoimmune diseases, thyroid diseases, and gallstones and bile duct stones, have been excluded.

Reviewer #1

This study investigated the ALT plus HBV virologic parameters to correlate with liver pathologic activity. Authors utilized statistical methods to define high/low HBV replication and to examine the accuracy of ALT/HBV virologic parameters in predicting liver pathology injury. Authors concluded that Quantitative HBsAg instead of HBV DNA is valuable in predicting significant hepatitis activity of HBeAg-positive CHB

Point 1: The manuscript is not reader friendly. Although the analysis or results are predictable, authors seem to describe or analyze the data in complex manner.

Response 1: Please see the penultimate paragraph of "Introduction" as well as the questions 5 to 7 of "Appendix" of the original manuscript.

  • Almost all previous studies refer to the existing natural history phase standards covering HBV DNA, which will lead to the actual importance of HBsAg underestimated because of the inherent correlation between HBV DNA and HBsAg. The importance of evaluating HBsAg should not be based on the natural history phase standards covering HBV DNA.
  • Almost all previous studies have HBeAg-positive patients as a whole, which will also lead to the actual importance of HBsAg underestimated because of the levels of HBV replication that affect HBV antigen expression are not stratified. In the "non-significant hepatitis activity" phase and the "significant hepatitis activity" phase of HBeAg-positive patients, the evolution direction of different HBV antigens and HBV DNA related to the levels of "hepatitis activity" in different stratifications is not exactly the same.
  • How to "reasonably" evaluate the importance of HBsAg and HBV DNA in predicting the natural history phases, and how to "reasonably" determine the cutoffs of HBsAg and HBV DNA in predicting the natural history phases, is the key problems that almost all previous studies have not solved, which is the main purpose of our manuscript.

Point 2: Table 2-4 are not well described in text.

Response 2: We may miss some content related to tables 2 to 4, which must be described in the text. However, we don't understand what not mentioned in the tables must be added to the text.

Point 3: A concern in the application of ALT is that the fluctuated features make the single measurement of ALT may compromise the analyzed results. Repeated measurement of ALT or analysis based on histology results, not including ALT, may be a more proper way.

Response 3: Please see the third paragraph of “Introduction”, table 4 of “Results” and the fifth paragraph of “Discussion”, as well as question 8 of “Appendix” of the original manuscript.

  • Liver pathological status is the best standard (not the gold standard) for determining “significant hepatitis activity”. However, a few patients with “significant hepatitis activity” always show mild liver necro-inflammation and slight liver fibrosis. In addition, there are sample errors as well as inter-observer and intra-observer bias in the evaluation of liver pathological status. Practically, repeated liver pathological examination is unrealistic.
  • ALT is a sensitive index to predict “significant hepatitis activity”. Theoretically, the abnormal increase of ALT levels indicates “significant hepatitis activity”. The fluctuation of ALT in patients with chronic HBV infection can be divided into fluctuation below the upper limit of normal reference and fluctuation above the upper limit of normal reference. Only the fluctuation above the upper limit of normal reference can indicate “significant hepatitis activity”, and only the fluctuation persistently below the upper limit of normal reference can indicate “non-significant hepatitis activity”. Practically, the upper limit of normal reference of ALT is still controversial, and the standard of upper limit of normal reference of ALT has not been unified, which has brought confusion to clinical practice and research. The results of previous studies and this study show that, even if ALT < 20 IU / L, about 10% of patients have extensive liver fibrosis (higher than or equal to S3) and about 5% of patients have liver cirrhosis (higher than or equal to S4). Therefore, it is inaccurate to predict “significant hepatitis activity” only based on the dynamic changes of ALT.
  • Accurate prediction of “significant hepatitis activity” also requires reference to other noninvasive parameters. Other noninvasive parameters that predict “significant hepatitis activity” in current clinical guidelines include at least HBV DNA and HBsAg. However, the rational cutoff for HBV DNA in predicting HBeAg-positive “significant hepatitis activity” has not been unified yet. Although EASL 2017 guidelines mentioned that HBsAg levels can be used as the basis for the phases of chronic HBV infection, but did not gave specific “quantization standard”.

Point 4: Prior studies revealed that a lower HBsAg is associated with advanced fibrosis in eAg+ CHB and also influenced by genotype of HBV. How about the impact of genotype on the study results?

Response 4: This is an issue that may have clinical importance. Unfortunately, this study is a retrospective cross-sectional study. Most patients lack HBV genotype data, which may be a flaw of this study. We look forward to future studies that have a similar design to this study, including HBV genotypes into analyses, to obtain more refined research results.

  • Previous studies based on the overall HBeAg-positive patients have reached a consistent conclusion: lower HBsAg has some predictive value for advanced fibrosis or cirrhosis, but whether advanced fibrosis or cirrhosis is more related to HBV genotype remains uncertain.
  • With the current availability of antiviral drugs, it may be more reasonable to manage patients based on the natural history phases than based on the stages of fibrosis. Therefore, exploring and evaluating the effective parameters that delimit the phases of natural history may be more clinically important.
  • Previous studies based on the natural history phase criteria covering HBVDNA and based on the overall HBeAg-positive patients showed that, HBsAg quantification "may be" related to HBV genotype. In other words, the correlation between HBsAg quantification and HBV genotype has not yet been determined.
  • Most of the subjects of this study are from East China, and the HBV genotypes of most of the subjects are genotype B and genotype C.

Point 5: In figure 2, why only used ALT stratification in ROC curve analysis? It needs clarification.

Response 5: Please see the question 8 of "Appendix" of the original manuscript.

  • The patients in HBeAg-positive "non-significant hepatitis activity" phase should be a highly homogenized population, whose characteristic must be sustained and stable "mild hepatic necro-inflammation, and no or slight hepatic fibrosis, and normal ALT". Therefore, the demarcation standard of HBeAg-positive "significant hepatitis activity" phase should be that "pathological grade higher than G1, or the pathological stage higher than S1, or ALT higher than or equal to ULNs".
  • On the contrary, the patients in HBeAg-positive "significant hepatitis activity" phase are a highly heterogenized population, whose behavior may be persistent or recurrent "significant hepatitis activity". In patients with early “significant hepatitis activity”, the pathological grade of higher than G1 or the stage of higher than S1 does not necessarily occur; in patients with late “significant hepatitis activity”, the high levels of pathological grade and stage may return to G1 and S1. In addition, in a few patients with “significant hepatitis activity”, the pathological grade may be never higher than G1 and the stage may be never higher than S1. Therefore, in the population separated from the "long term stable non significant hepatitis activity", i.e. the population entered “significant hepatitis activity” phase, further identifying the subpopulation who are still active, should only refer to the standards of " ALT higher than or equal to ULNs".

Point 6: Baseline characteristics should be supplemented in Table 1, such as BW or BMI, comorbid diseases, or even steatosis grade in liver histology, etc. Some of them may have impact on the presentation of ALT level.

Response 6: Please see the description of the “Study population” of “Materials and Methods” of the original manuscript.

Some comorbid diseases, including coinfection with other hepatotropic viruses, Epstein-Barr virus, cytomegalovirus; Schistosomiasis japonica liver disease, nonalcoholic/metabolic fatty liver disease, drug-induced liver injury, excessive drinking, autoimmune diseases, thyroid diseases, and gallstones and bile duct stones, have been excluded.

Reviewer 2 Report

In the study Yueping Ding et all titled “Quantitative HBsAg versus HBV DNA in predicting significant 2 hepatitis activity of HBeAg-positive chronic HBV infection” authors found that quantification of HBsAg may be important as a predictor of significant hepatitis activity in patients chronically infected with HBV HBeAg-positive.

The aim of the study is completely clear. The results of this study has some novelty to previous literatures regarding the relationship between amount HBsAg and prognosis of natural course of HBV infection. The authors use univariate but not multivariate analyzes to assess the predictive value of HBsAg in predicting significant hepatitis activity. The authors used both standard and non-standard statistical methods to define thresholds that define high- and low-replication groups of the virus and to detection of diagnostic utility. Of course, it is debatable to what extent the established thresholds will be universal for other groups of HBeAg positive patients, but for the group of patients analyzed in this manuscript, the legitimacy of determining these thresholds cannot be denied.

 It is a study with large amount of cases to final conclusions, and will be helpful for further research in this field among specific subpopulation.

The paper makes original contribution and it is clinically exhaustive. The manuscript is well written, seems accurate and well organized. The authors clearly presented any doubts concerning the investigation conducted by them.

As a reader interested in the topic in question, I would like to see the combined effect of the analyzed variables on the prediction of individual stages of liver activity, e.g. ROC curves created by combining the values for HBsAg and HBeAg variables in the subgroup of patients with possible high HBV replication and for the variables HbeAg and HBV DNA in patients with possible low HBV replication. The analysis of several factors can significantly increase the possibility of predicting the next stages of chronic HBV infection, however, on the other hand, the purpose of the work presented in the manuscript was fulfilled.

I noticed a few minor editing errors in the work, therefore I ask you to double-check the text of the manuscript. 

Author Response

Reviewer #2

In the study Zhanqing Zhang Yueping Ding et all titled “Quantitative HBsAg versus HBV DNA in predicting significant hepatitis activity of HBeAg-positive chronic HBV infection” authors found that quantification of HBsAg may be important as a predictor of significant hepatitis activity in patients chronically infected with HBV HBeAg-positive.

Point 1: The aim of the study is completely clear. The results of this study has some novelty to previous literatures regarding the relationship between amount HBsAg and prognosis of natural course of HBV infection. The authors use univariate but not multivariate analyzes to assess the predictive value of HBsAg in predicting significant hepatitis activity. The authors used both standard and non-standard statistical methods to define thresholds that define high- and low-replication groups of the virus and to detection of diagnostic utility. Of course, it is debatable to what extent the established thresholds will be universal for other groups of HBeAg positive patients, but for the group of patients analyzed in this manuscript, the legitimacy of determining these thresholds cannot be denied.

Point 2: It is a study with large amount of cases to final conclusions, and will be helpful for further research in this field among specific subpopulation.

Point 3: The paper makes original contribution and it is clinically exhaustive. The manuscript is well written, seems accurate and well organized. The authors clearly presented any doubts concerning the investigation conducted by them.

Point 4: As a reader interested in the topic in question, I would like to see the combined effect of the analyzed variables on the prediction of individual stages of liver activity, e.g. ROC curves created by combining the values for HBsAg and HBeAg variables in the subgroup of patients with possible high HBV replication and for the variables HbeAg and HBV DNA in patients with possible low HBV replication. The analysis of several factors can significantly increase the possibility of predicting the next stages of chronic HBV infection, however, on the other hand, the purpose of the work presented in the manuscript was fulfilled.

Response 1-4: Thank review #2 for reading our manuscript carefully and fully understanding the importance of our work. We admire review #2 deep and comprehensive understanding of the background of this study.

Point 5: I noticed a few minor editing errors in the work, therefore I ask you to double-check the text of the manuscript.

Response 5: We carefully reviewed the original manuscript and found 4 editing errors, as follows:

  • Abbreviations of Table 1.: ……Stage, pathological stage Stage
  • Figure legend of Figure 2.: ……A, C and E: , ……; B, D and F: ,
  • The first paragraph of Page 9: In overall population, the The……
  • Discussion: guideline→guidelines

Round 2

Reviewer 1 Report

I have no further comments on this manuscript